# Flattening the Curve of Flexible Space Robotics

Timothy Sands 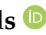

Sibley School of Mechanical and Aerospace Engineering, Cornell University, Ithaca, NY 14853, USA;
tas297@cornell.edu

**Abstract:** Infrastructure monitoring, inspection, repair, and replacement in space is crucial for continued usage and safety, yet it is expensive, time-consuming, and technically very challenging. New robotics technologies and artificial intelligence algorithms are potentially novel approaches that may alleviate such demanding operations using existing or novel sensing technologies. Space structures must necessarily be very light weight due to the high costs of placing robots in space. Several methods are proposed and compared to control highly flexible space robotics, where a key challenge is the presence of flexible resonant modes at frequencies so low as to reside inside typical feedback controller bandwidths. Such conditions imply the very action of sending control signals to the ultra-light weight robotics will cause structural resonance. Implementations of incrementally increasing order are offered, achieving an over ninety percent performance improvement in trajectory tracking errors, while improvement using unshaped methods merely achieve a twenty-four percent improvement in direct comparison (where the only modification is the proposed control methodology). Based on superior performance, single-sinusoidal trajectory shaping is recommended, with a corollary benefit of preparing future research into applying deterministic artificial intelligence whose current instantiation relies on single-sinusoidal, autonomous trajectory generation.

**Keywords:** robotics; monitoring; repair; replace; infrastructure; space robotics; flexible; structural-controls interaction; trajectory shaping

## 1. Introduction

This article describes current novel methods in space robotics in the context of former methods, inspiring paths for future ground-breaking discovery in the maintenance of on-orbit space infrastructure. The modern state of the art is compared to classical methods and recently proposed methods; furthermore, future novel methods are inspired and grounded in a basis of lessons learned in the comparisons of classical and modern methods.

### 1.1. Broad Context and Highlighting Importance

Currently, placing material into low earth orbit costs roughly ten thousand U.S. dollars per pound [1–3]. One key lifetime limitation is the finite quantity of fuel brought to orbit [3]. Other lifetime limitations include the gradual degradation of solar panels, reducing electrical power generation, and failure of other spacecraft components (e.g., electrical devices such as radios, the gradual decay of moving parts such as gyroscope bearings, etc.) [4]. An attractive remedy that is currently an active area of research includes in situ resource replenishment and requisite disassembly and assembly (construction), as depicted in Figure 1a [5]. Benchboard spacecraft robotic laboratories (in Figure 1b) [6] are useful to validate advanced thinking on the basic scientific aspects of the possibilities of future operations establishing a functional supply chain of logistical support to spacecraft on orbit.

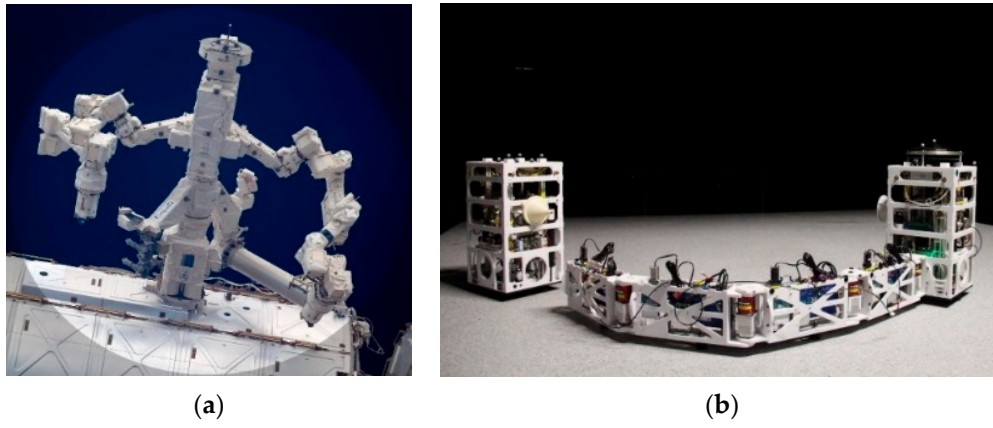

(**a**)

(**b**)

**Figure 1.** (**a**) NASA's remote-controlled robotic refueling mission on orbit [1]. (**b**) U.S. Naval Postgraduate School's autonomous robotic laboratory free-floating simulator on a planar air-bearing table [2].

### 1.1.1. Modeling

Modeling space robots (such as those depicted in Figure 2) from first principles articulated by Chasle [7] begins with application of Newton's second law of motion [8] for translation and Euler's moment equations [9] for rotation. Utilizing the finite element method, the robot may be discretized into a chosen number of nodes for application of these first principles, establishing a system of differential equations whose order is driven by the chosen number of nodes.

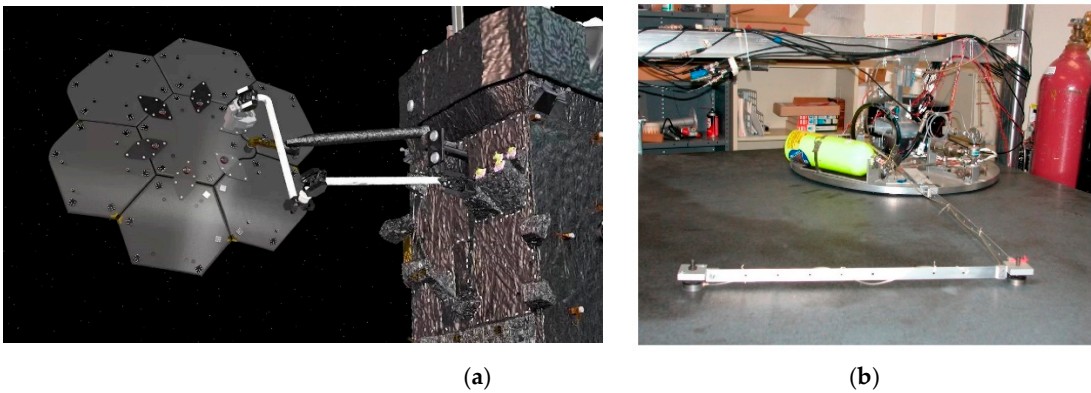

(**a**)

(**b**)

**Figure 2.** Robotic assembly and manufacturing in space. (**a**) NASA and Maxar Technologies SPIDER (planned) robotic assembly in space [5]. (**b**) Autonomous robotic laboratory free-floating simulator on a planar air-bearing table [6]. This testbed is the subject of modeling and control efforts, leading to the discovery of whiplash compensation of flexible space robotics [6].

### 1.1.2. Classical Methods

Traditionally, the nonlinear coupled motion terms in both translation and rotation equations are neglected, linearized, or simplified by assumptions, permitting classical treatment of linear, time-invariant methods [10]. The treatment begins with gain stabilization through feedback and may be augmented with frequency-dependent filtering to shape the frequency-response of the robotic control attempting to negate the deleterious effects of structural flexibility.

### 1.1.3. Modern Methods

The term "modern" is often used to either indicate problem formulation in space-variable form, also called "state space" form [11,12], or, alternatively, deterministic optimization (as opposed to stochastic optimization commonplace in classical methods) [13].

Modern methods, even optimal instantiations, ubiquitously impose a specified limited form of the robotic control equation. Typically, the control is forced to be a negative feedback or errors, where gains associated with each error are discerned to optimize some specified performance function. One very common form of a modern optimal feedback controller is the linear quadratic regulator [14] (amongst others, including robust L1 adaptive control [15,16]). An alternative modern method eliminated the mandatory use of feedback to discern a feedforward (open loop) approach to control flexible robotic systems where the deleterious responses of flexible, multi-body motion are codified in a method called input shaping [17,18], where the anticipated strain energy is distributed with time to create a time-delayed control approach for flexible space robotics [19]. The methods (feedforward and feedback) have also been combined [20].

### 1.1.4. Recently Proposed Methods

Post-modern methods eliminate the assertion of the presumed negative feedback form of the control with variable gains. The differential equations mentioned in Section 1.1.1 may be expressed in vector-matrix form, establishing dynamic constraints to form a time-varying, nonlinear Hamiltonian system [21] that may be optimized in accordance with the methods of Pontryagin [22]. The result of this approach applied to highly flexible space robotics is the recently revealed whiplash compensation of flexible space robotics [23]. The method was developed on a university benchboard free-floating space robot simulator, depicted in Figure 2b, representing high-end missions such as the NASA/Maxar robotic assembly mission (depicted in Figure 2a) currently under construction. Whiplash compensation, as proposed, is an open-loop optimal approach (akin input shaping) focused on shaping the control to optimize the rotational motion of highly flexible space robots. The unexpected counter-steering devised inspired renewed focus on shaping desired trajectories in the context of optimized controls, leading to the just proposed deterministic artificial intelligence [24].

### 1.1.5. Potential Methods

The open-loop optimal shaping of the commanded trajectory in addition to the unexpected results of the nonlinear constrained optimization embodied in whiplash compensation inspire study of alternative trajectory shaping options. This article will propose sinusoidal trajectory shaping options [25] to provide methods to steer the control signal's frequency content away from problematic frequencies. This investigation is inspired by the very recently proposed deterministic artificial intelligence method [24] utilizing sinusoidally shaped input trajectories. Especially since the newly proposed method is parameterized in terms of time-variant mass and mass moment of inertia [26,27], the potential for fruitful application to autonomous space robotics seems high since the application requires operations amidst sudden increases and decreases in mass and mass moments. This potential is proposed for future research, permitting deterministic artificial intelligence to be applied to highly flexible space robotics.

### 1.2. Statements of Novelty

This paragraph succinctly states the novel contributions offered in the article.

1. Trade-off study: compensate only for purposes of insuring stability versus compensation to flatten the frequency response curve, enhancing regular predictability with regards to frequency of excitation.
2. Direct comparison of classical methods and a smoothed version of recently proposed whiplash compensation (originally proposed with non-asymptotically smooth ramp trajectories) [23].
3. Additional direct comparison with the newly proposed single-sinusoidal trajectory limitation currently the baseline approach for deterministic artificial intelligence methods.
4. Elaboration of future research using recently proposed deterministic artificial utilization of (autonomously generated) a single-sinusoidal trajectory.

### 1.2.1. Trade-Off Study on Application of Classical Methods

Classical proportional plus integral plus derivative control is designed to satisfy the nominal performance specifications (e.g., rise time, overshoot, settling time, etc.). Following the well-documented methods of flexible system compensation. Second-order filters are sequentially applied to each flexible mode: notch filtering of resonant response and bandpass filtering of anti-resonant responses associated with free-free vibrations of on-orbit space robots. Classical procedures primarily focus on portions of flexible modes (either resonance or anti-resonance) that by generation of cross-over frequencies. Accordingly, only a single notch or bandpass filter is commonly used, which also helps limit system order, since addition of second-order filters in the feedback path increases the closed-loop system order. Trade-off studies are performed, evaluating both the performance and increase in system order for progressively more structural filtering, eventually seeking to "flatten the curve" (the magnitude response plots), essentially nullifying the flexible affects. While a predictable response can be tuned to the desired performance specification, these classical methods suffer from limited robustness to variation in system natural frequencies. Iterative application of classical structural filtering, as instantiated in this research, realizes an eleven to four hundred and twenty percent improvement in the gain margin and minus five to sixty-four percent improvement in the phase margin, with an up to twenty-four percent improvement in tracking error in response to step commands. On the downside, the methods increase the system order up to fifty, implying a high increase in computational burden.

### 1.2.2. Trade-Off Study of Modern Simple-Trajectory Shaping Methods and Post-Modern Whiplash Compensation

Trajectory shaping (input shaping, adaptive input shaping, zero-vibration, and zero-vibration derivative shaping, etc.) are recently proposed methods seeking to mitigate the deleterious effects of step inputs that theoretically exhibit an infinite bandwidth [28,29]. While predictable response can also be tuned to the desired performance specification (as with classical structural filtering), these methods also suffer from limited robustness to variation in system natural frequencies and are mentioned here contextually, but not included in the analytic treatment. Most recently, whiplash compensation has been proposed in the literature [23], and a direct comparison is made in this article.

### 1.2.3. Proposed Trajectory Generation

The aforementioned techniques use system analysis to identify the natural frequencies with modeling, and then design the controls based on the identified frequencies inherently limiting robustness when the natural frequencies are known, poorly modeled, or time variant.

*An alternative paradigm is proposed here, instead designing controls based on a single frequency stemming from the desired speed of the response, and then ensuring that the single frequency is well outside the possible proximity of the system's natur.*

The proposed method accepts step commands by autonomously converting the commands into state trajectories based on single frequency sinusoids, as proposed in recent publications on deterministic artificial intelligence [24,25,27]. The proposed methods are implemented on four systems of incrementally increasing order, achieving an over ninety percent performance improvement in trajectory tracking errors with step inputs, while improvement using unshaped methods merely achieve a twenty-four percent improvement in direct comparison (where the only modification is application of the proposed methodology).

### 1.3. Main Aims of the Article

The goal is to lay the groundwork from classical methods through to modern and post-modern methods, to inspire and recommend future basic scientific research in autonomous space robotics, permitting a single robotic inspection, replenishment, and assembly robot system to service satellites with very disparate connections (Figure 3) that were not designed

for in situ replenishment and assembly operations, as depicted in Figure 4's remotely controlled prototype for the suggested autonomous system.

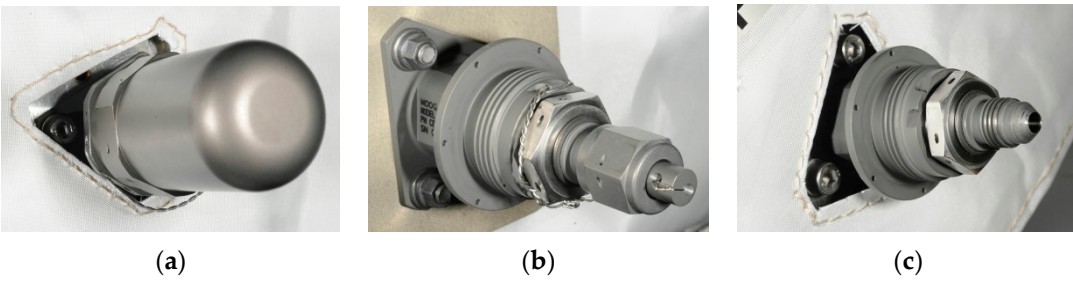

(**a**)      (**b**)      (**c**)

**Figure 3.** Disparate satellite fuel valves, illustrating a challenging aspect of refueling an arbitrary spacecraft on orbit: (**a**) tertiary cap with "lock wire" visible underneath; (**b**) safety cap/actuation nut with securing lock wire; (**c**) exposed fuel valve [3].

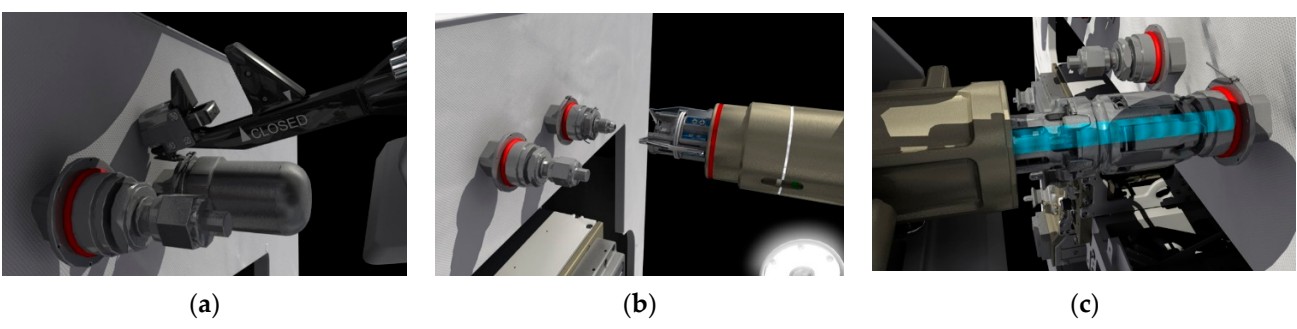

(**a**)      (**b**)      (**c**)

**Figure 4.** Operations required to connect to various satellite fuel valves, illustrating a challenging aspect of refueling an arbitrary spacecraft on orbit [3]: (**a**) start with wire cutting; (**b**) connect nozzle tool with hose and transfer fuel; (**c**) leave behind quick-disconnect when re-fueling complete (for future refueling operations).

## 2. Materials and Methods

Modeling and modal system identification were followed by focusing on the development of the decision and control criteria, development and shaping or the desired (commended) input trajectory, and combinations of both shaping and control developments. This foundation leads to an instinct that the parameterization methods that include time-varying mass and mass moment of inertia can be developed to mitigate rapid increases and decreases associated with grasping and releasing target-spacecraft of unknown or assumed poorly known mass properties. The suggested path for future basic scientific research is deterministic artificial intelligence recently developed for autonomous unmanned underwater vehicles [24].

Modeling is illustrated to be a combination of mathematical modeling [30,31] of rigid bodies [32,33] foremost followed by augmentation of the rigid body model with flexible components (e.g., robotic manipulator arms) [11,34,35]. Together, the combined model will establish the flexible space robot depicted in the SIMULINK model in Figure 5a. The "manual control selection" box contains the manual routing switches, permitting direct comparisons, where the only variation is the method of controlling the highly flexible space robot. Analysis performed in this manner produce figures of merit for decision makers deciding which approach to use for autonomous control in a given situation. The manual switches allow iteration of the various approaches coded in the boxes to the left of Figure 5a: PID, structural filtering, selectable feedforwards, and selectable (shaped) commanded trajectories. Figure 1b illustrates the contents of the flexible space robot subsystem in the far right-hand side of Figure 5a,b.

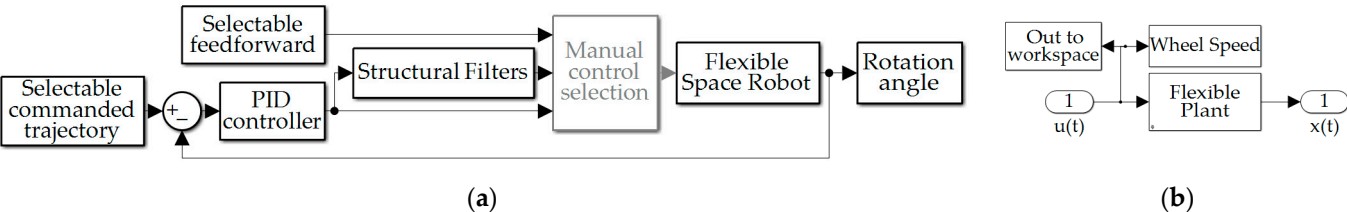

**Figure 5.** (**a**) Simulations created in SIMULINK, to compare the disparate approaches. (**b**) Flexible space robot subsystem depicted in (**a**).

### 2.1. Modeling and Analysis of Flexible Space Robots

Modeling highly flexible space robots begins with treatment of the rigid bodies, which subsequently have appended to them highly flexible robotic appendages, where maneuvering the body is accomplished by a spinning wheel [36] exchanging angular momentum with the body [37]. The addition of the flexible robotic appendages is accomplished by repeated application of rigid body treatments at a chosen number of nodes, where the nodes result from discretization of the continuous structure of the robotic appendage.

### 2.1.1. Rigid Body Dynamics

*Michel Chasles's theorem [7] allows us to simply invoke Newton [8] and Euler's equations [9] to fully describe the six degrees of freedom of mechanical motion [38].*

These relationships were very recently used together with Pontryagin's treatment of optimization [22] to reveal whiplash compensation for the flexible space robot depicted in Figure 5b [23], in addition to the development last year of the burgeoning field of deterministic optimization [24] that uses these relationships to formulate self-awareness statements that are augmented by either nonlinear adaptive feedback or optimum (in the 2-norm sense) learning, using a reparameterization of the same basic dynamic relationships.

The basic relationships relating force to acceleration (per Newton) and torques to angular acceleration (per Euler) are illustrated in Equations (1) and (2), respectively, where the quantities are measured relative to a non-rotating reference frame (called "inertial"), while their expression in coordinates of the rotating reference frames are expressed in Equations (3) and (4), respectively. Simply invoking the two relationships comprise Chasle's theorem.

$$F|_{inertial} = ma|_{inertial} = m\dot{v}|_{inertial} = m\ddot{x}|_{inertial} \tag{1}$$

$$\tau_{inertial} = J\alpha|_{inertial} = m\dot{\omega}|_{inertial} \tag{2}$$

$$F = m\ddot{x} + m\frac{d\omega}{dt} \times r\prime + 2m\omega \times v\prime + m\omega \times (\omega \times r\prime) \tag{3}$$

$$\tau = J\dot{\omega} + \omega \times J\omega \tag{4}$$

Equations (3) and (4) are implemented in this article as the rigid body mode of a flexible space robot designed for in situ replenishment or assembly operations on orbit where:

$F$ is the vector sum of the external forces acting on the body;
$\tau$ is the vector sum of the external torques acting on the body;
$\omega$ is the angular velocity;
$v\prime$ is the velocity relative to the rotating reference frames;
$r\prime$ is the position vector of the object relative to the rotating refence frame;
$a\prime$ is the acceleration relative to the rotating reference frame;
$m\frac{d\omega}{dt} \times r$ is the Euler acceleration;
$2m\omega \times v\prime$ is the Coriolis acceleration;
$m\omega \times (\omega \times r\prime)$ is the centrifugal acceleration;
$J$ is the mass moments of inertia.

Section 2.1.1 indicates the same results were obtained by Hamilton. Hamilton's principle implies that the Lagrange method must be as follows:

$$\text{Minimize } Cost[x(g), u(g), t] = \int_{t_0}^{t_f} L(\dot{x}, x)dt \tag{5}$$

$$\text{Subject to } \quad \dot{x} = u \tag{6}$$

where

$$L = Kinetics\ energy - potential\ energy = \frac{1}{2}mV^2 + mgh = \frac{1}{2}m\dot{x}^2 + mgh \tag{7}$$

$$\text{Writing the control Hamiltonian: } H(\lambda, x, u) \equiv L(x, u) + \lambda u \tag{8}$$

$$\text{Pontryagin's minimization condition: } \frac{\partial H(\lambda, x, u)}{\partial u} = \frac{\partial L(x, u)}{\partial u} + \lambda = 0 \tag{9}$$

Differentiating leads to a minimization equation expression that may be combined with the Adjoint equation to produce the Euler–Lagrange equation, which by algebraic manipulation becomes Newton's Law.

$$\frac{d}{dt}\left(\frac{\partial L(x, u)}{\partial u} + \lambda\right) = \frac{d(0)}{dt} = 0 \rightarrow \underbrace{\frac{d}{dt}\left(\frac{\partial L(x, u)}{\partial u}\right) - \frac{\partial L(x, u)}{\partial x} = 0}_{Euler-Lagrange\ equation} \tag{10}$$

$$\frac{d}{dt}\left(\frac{\partial L(x,u)}{\partial u}\right) - \frac{\partial L(x,u)}{\partial x} = \frac{d}{dt}\left(\frac{\partial L(x,u)}{\partial u}\right) - \frac{\partial L(x,u)}{\partial x} = \frac{d}{dt}\left(m\dot{x}\right) - \frac{\partial L(x,u)}{\partial x}$$
$$= m\ddot{x} - \frac{\partial L(x,u)}{\partial x} = 0 \tag{11}$$

$$\underbrace{\frac{\partial L(x, u)}{\partial x}}_{F} = m\underbrace{\ddot{x}}_{A} \Leftrightarrow \underbrace{F = mA}_{Newton's\ Law} \tag{12}$$

where:

$H(\lambda, x, u)$ is the Hamiltonian;
$L(x, u)$ is the Lagrangian;
Kinetic energy is a function of velocity;
Potential energy is a function of displacement;
$\lambda$ is the nomenclature for the adjoints or co-states associated with each state.

Hamilton's principle of minimizing the Lagrangian results in two methods to derive the equations of motion: Newton's Law and the Euler–Lagrange Equation both represented by the so-called "double-integrator" (Equations (1) and (2)) initially until elaborated by expression in rotating reference frames (Equations (3) and (4)).

The three degrees of freedom of translational motion represented in Equations (1) and (3) together with the three degrees of freedom of motion of rotation in Equations (2) and (4) together completely specify six degrees of freedom ("6-DOF") of mechanical motion. The cross products of motion in Equations (3) and (4) create a high level of complication, such that a general solution of this differential equation without simplifying assumptions remains unsolved.

Assuming a feedback control whose signal is restricted to gains, $K$ multiplied by the state errors, controlling the rigid body equations of motion, begin with the so-called double integrator relationships for both translation and rotation, and are often represented by the spring–mass–damper system depicted in Figure 6a, whose response magnitude and phase may be plotted on log-normal graphs, as depicted in Figure 6b. The presence of two integrators in the response function numerator dominates the low frequency response with a response reduction by forty decibels per decade of frequency up to the breakpoint designated by the Greek letter omega, $\omega$, after which the response reduces 20 decibels per decade.

A higher number of integrators in the denominator is called a "pole excess" and the high frequency phase lag asymptotically approaches this pole excess with increasing frequency.

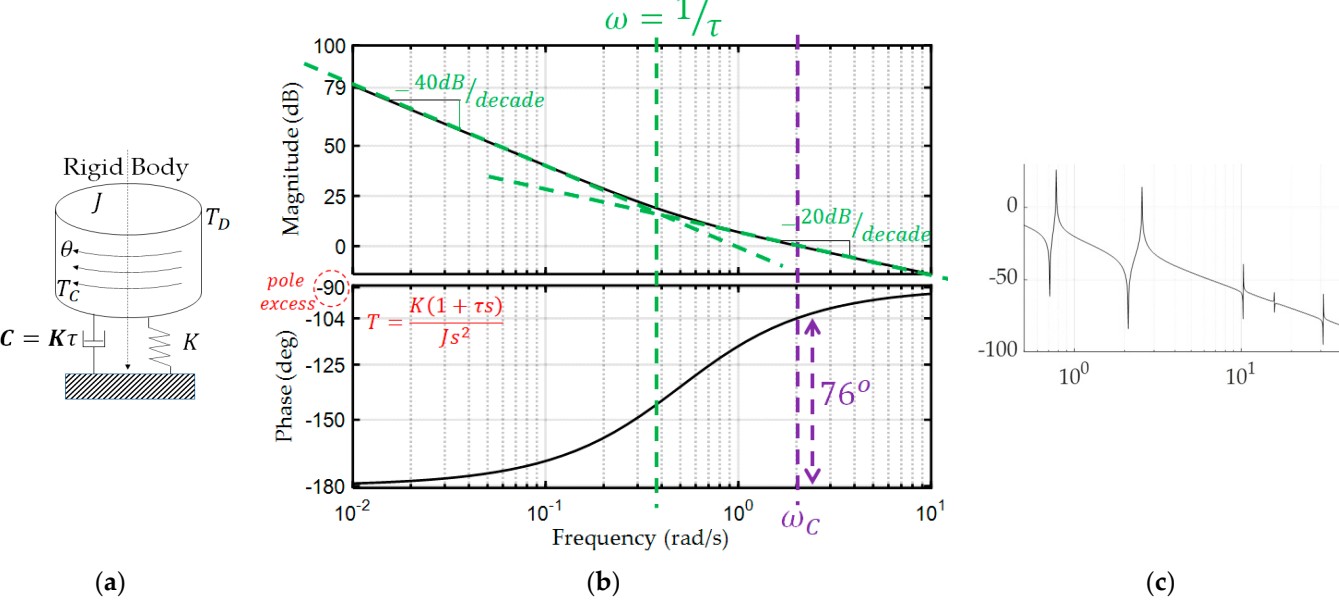

(a)

(b)

(c)

**Figure 6.** Modeling and analysis of rigid bodies with respect to frequencies. (**a**) Modeling rigid bodies as systems with springs, masses (and mass moments of inertia), and dampers. (**b**) Frequency response analysis of the rigid body portion (or "rigid body mode") of the flexible space robot. (**c**) Frequency response analysis of the flexible body portion of the space robot depicted in Figure 7b (note the five natural frequencies corresponding to the selection of five nodes, while the referenced system has also been modeled using eight nodes, as displayed in Figure 2 of [23]).

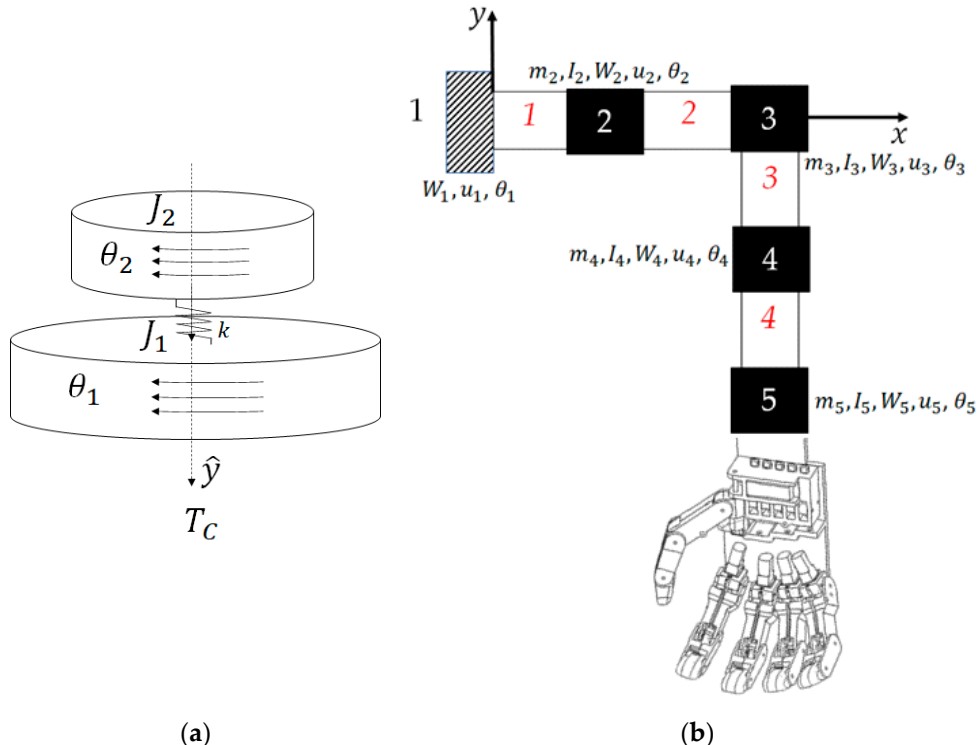

(a)

(b)

**Figure 7.** Modeling and analysis of the flexible bodies discretized into the four nodes depicted in (**b**) of the mass moment of inertias $I_2$ through $I_5$ appended to the rigid body depicted in (**a**) as $J_1$, where $J_2$ in (**a**) represents the sum of constituent inertias at nodes 2–5 in (**b**).

The seemingly simplified rigid body mode becomes the core of the flexible system in its labeling as the "rigid body mode" of the total flexible space robot. To complete the modeling effort, flexible parts are modeled separately and appended to the rigid body.

### 2.1.2. Flexible Body Dynamics

Discretizing the flexible appendages into a chosen number of nodes (four nodes used here illustratively, as depicted in Figure 7b) and rotational and translational equations from Section 2.1.1 are repeatedly applied at each of the chosen nodes resulting in a system of equations of dimension equal to the chosen number of nodes.

The system of equations expresses in three axes is parameterized in accordance with the single ($\hat{z}$-axis) rotation depicted in Figure 8a, representing the flexible space robot simulator depicted in Figure 8b. Notice the end effector manipulator position in this illustration is achieved by control of the attitude of the central rigid body. Omitted here also is the grasping action of the robotic gripper (depicted in Figure 7b). The rigid spacecraft hub in Figure 8 is parameterized in the reference frame (black font $\hat{x}, \hat{y}, \hat{z}$) centered at point frame $O$, which rotates an angle $\theta$ to angular positions designated by the red dashed line with a red font ($\hat{X}, \hat{Y}, \hat{Z}$), while the actuator wheel (purple font $\hat{x}, \hat{y}, \hat{z}$) is centered at point frame $O_W$, which rotates an angle $\theta_W$ to angular positions designated by the purple dashed lines with a purple font ($\hat{x}\prime, \hat{y}\prime, \hat{z}\prime$). The vector of multi-dimensional forces $F$ on the robotic arm are decomposed into radial $\bar{r}_F$ and normal $\bar{u}_F$ components.

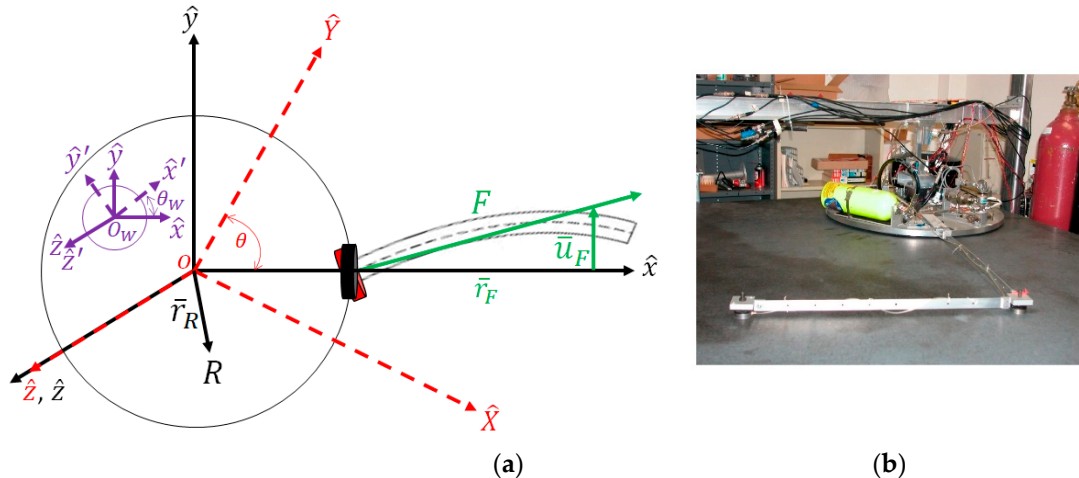

(**a**) (**b**)

**Figure 8.** Schematic representation (**a**) of the flexible space robot depicted in Figure 7, representing the free-floating flexible robot simulator at the U.S. Naval Postgraduate School depicted in (**b**).

Equations of Motion for the flexible system (rigid body plus flexible appendages may be conveniently derived using Euler–Lagrange's Method. Lagrange's method is appropriate regardless of various coordinate systems since it utilizes the systems kinetic and potential energy (V) in Equation (13) to derive the equations of motion.

$$Kinetic\ energy = \frac{1}{2} = \int_R (V_R \cdot V_R)dm + \int_F (V_F \cdot V_F)dm + \int_W (V_W \cdot V_w)dm \quad (13)$$

where:

$V_R = \dot{\theta}\hat{k} \times r_R$ is the velocity of a particle on the rigid body;

$V_F = \dot{\theta}\hat{k} \times r_F + \delta + \dot{\theta}\hat{k} \times \delta$ is the velocity of a particle on the flexible body;

$V_W = \left(\dot{\theta} + \dot{\theta}_W\right)\hat{k} \times r_W + \dot{\theta}\hat{k} \times r_o$ is velocity of a reaction wheel particle;

$r$ vectors are position vectors to the respective particles;

$\delta$ is elastic deformation vectors of particles on the flexible body;

$\hat{i}, \hat{j}$, and $\hat{k}$ are unit vectors in the $x', y', z'$ axes of the reaction wheel.

Expanding all the terms in the equation for kinetic energy produces Equation (14):

$$
\begin{aligned}
Kinetic\ energy = {} & \tfrac{1}{2}\int_R (x_R^2 + y_R^2)\,dm + \tfrac{1}{2}\int_F \left[\dot{\theta}^2(x_R^2 + y_R^2) + \left(\dot{\delta}_x^2 + \dot{\delta}_y^2\right) + \dot{\theta}^2\left(\delta_x^2 + \delta_y^2\right)\right] dm \\
& + \tfrac{1}{2}\int_F \left[2\dot{\theta}\left(x_f\dot{\delta}_y - y_f\dot{\delta}_x\right) + 2\dot{\theta}^2\left(x_f\delta_x + y_f\delta_y\right) + 2\dot{\theta}\left(\dot{\delta}_y\delta_x + \dot{\delta}_x\delta_y\right)\right] dm \\
& + \tfrac{1}{2}\int_W \left[\dot{\theta}^2(x_0^2 + y_0^2) + \dot{\theta}^2(x_w^2 + y_w^2) + 2\dot{\theta}^2(x_0 x_w + y_0 y_w) + \dot{\theta}_w^2(x_w^2 + y_w^2)\right] dm \\
& + \tfrac{1}{2}\int_W \left[2\dot{\theta}\dot{\theta}_w(x_w^2 + y_w^2) + 2\dot{\theta}_w\dot{\theta}(x_0 x_w + y_0 y_w)\right] dm
\end{aligned}
\tag{14}
$$

Utilizing the generalized coordinates $\theta, \theta_W, \delta$, assuming the reaction wheel rotates about its center of mass, and invoking the small displacement assumption, simplifies the kinetic energy equation to Equation (15):

$$
\begin{aligned}
Kinetic\ energy \quad = {} & \tfrac{1}{2}I_{zz}\dot{\theta}^2 + \tfrac{1}{2}I_w\dot{\theta}_w^2 + \tfrac{1}{2}I_w\dot{\theta}\dot{\theta}_w + \tfrac{1}{2}\int_F \left(\dot{\delta}_x^2 + \dot{\delta}_y^2\right) dm + \dot{\theta}\int_F \left(\dot{\delta}_x^2 + \dot{\delta}_y^2\right) dm \\
& + \dot{\theta}\int_F \left(x_F\dot{\delta}_y + y_F\dot{\delta}_x\right) dm
\end{aligned}
\tag{15}
$$

where the mass moment of inertia with respect to the $\hat{z}$ axis is in Equation (16).

$$
I_{zz} = I_{zz}^R + I_{zz}^F + I_{zz}^w = \int_R \left(x_R^2 + y_R^2\right) dm + \int_R \left(x_F^2 + y_F^2\right) dm + \int_R dm + m_w\left(x_0^2 + y_0^2\right) \tag{16}
$$

### 2.2. Modal System Identification

Convert the elastic deformation $\delta$ into cantilever modal coordinates [12] using $\delta_x = \sum_{i=1}^n \phi_i^x q_i(t)$ and $\delta_y = \sum_{i=1}^n \phi_i^y q_i(t)$. This relationship describes that the displacement at time t of a point at position x along the FSS is related to the mode shape $\phi$ (x and y components) and the generalized coordinate or modal coordinate, $q_i(t)$, a sinusoidal function. The mode shape is a time-independent characteristic of the flexible structure. Using this relationship, kinetic energy $T$ in terms of the generalized coordinates is Equation (17).

$$
\tau = \frac{1}{2}I_{zz}\dot{\theta}^2 + \frac{1}{2}I_w\dot{\theta}_w^2 + \frac{1}{2}I_w\dot{\theta}\dot{\theta}_w + \frac{1}{2}\int_F \left(\sum_{i=1}^n\sum_{i=1}^n\left[\phi_i^x\phi_j^x + \phi_i^y\phi_j^y\right]\dot{q}_i\dot{q}_j\right) dm + \dot{\theta}\int_F \left(x_f\sum_{i=1}^n\phi_i^y\dot{q}_i - y_f\sum_{i=1}^n\phi_i^x\dot{q}_i - \right) dm \tag{17}
$$

$$
\tau = \frac{1}{2}I_{zz}\dot{\theta}^2 + \frac{1}{2}I_w\dot{\theta}_w^2 + \frac{1}{2}I_w\dot{\theta}\dot{\theta}_w + \frac{1}{2}\sum_{i=1}^n\dot{q}_i^2 + \dot{\theta}\sum_{i=1}^n D_i\dot{q}_i^2 \tag{18}
$$

where rigid elastic coupling term $D_i = \int_F \left(x_f\phi_i^y - y_f\phi_i^x\right) dm$.

Potential energy comes from stiffness and can be expressed in generalized coordinates as $V = \frac{1}{2}\sum_{i=1}^n \omega_i^2 q_i^2$, where $\omega_i$ is the natural frequency of the $i$th mode. The virtual work is given [11] as $\delta W = \sum T\delta(\theta + \theta_w) - \sum T\delta\theta + T_D\delta\theta = \sum T\delta\theta_w + T_D\delta\theta$ where $T_D$ is disturbance torques and $\sum T$ is total torques (not to be confused with T = kinetic energy). Revisiting the Euler–Lagrange equation, substituting potential and kinetic energy yields the equations of motion in terms of the generalized coordinates.

$$
\sum \tau = I_{zz}\ddot{\theta} + \sum_{i=1}^n D_i\ddot{q}_i \tag{19}
$$

$$
\ddot{q}_i + 2\xi\omega_i\dot{q}_i + \omega_i^2 q_i + D_i\ddot{\theta} = 0 \tag{20}
$$

Notice Newton's law results in the identical equations of motion accounting for the stiffness between the adjacent nodes of the flexible robot appendage.

Newton's Law: $\sum F = ma$ applies at each node of the system, where the coordinates are defined in the hybrid-coordinate system [12].

$$m_1 \ddot{x}_1 = -k_1 x_1 + k_2 (x_2 - x_1) \tag{21}$$

$$m_2 \ddot{x}_2 = -k_2 (x_2 - x_1) + F \tag{22}$$

$$\underbrace{\begin{bmatrix} m_1 & 0 \\ 0 & m_2 \end{bmatrix}}_{[M]} \underbrace{\left\{ \begin{matrix} \ddot{x}_1 \\ \ddot{x}_2 \end{matrix} \right\}}_{\ddot{x}} + \underbrace{\begin{bmatrix} k_1 + k_2 & -k_2 \\ -k_2 & k_2 \end{bmatrix}}_{[K]} \underbrace{\left\{ \begin{matrix} x_1 \\ x_2 \end{matrix} \right\}}_{x} = \underbrace{\left\{ \begin{matrix} 0 \\ 1 \end{matrix} \right\} F}_{\{F\}} \tag{23}$$

$[M]$ is the system's global mass matrix (to be derived by finite element analysis);
$[K]$ is the system's global stiffness matrix (to be derived by finite element analysis);
$\{F\}$ is the force vector.

Including structural damping results in $[M]\{\ddot{x}\} + [C]\{\dot{x}\} + [K]\{x\} = \{F\}$. The two degrees of freedom are x/y displacements and $\theta$ rotations. x/y rotations are mutually exclusive since we are not allowing plane stress and plane strain; that is, the members are not allowed to lengthen/shorten (bending displacements only). So, for the horizontal members, the $x_1$ in the equation of Newton's Law corresponds to the Y direction bending displacements. Similarly, for vertical members, the $x_1$ corresponds to x-direction bending displacements. The nodal degrees of freedom are constrained to zero at the attachment point of the flexible and rigid bodies (forcing the appendage to stay attached). Note also that the corner node of the flexible body is permitted, with no x or y displacement degrees of freedom, leaving only $\theta$ rotations. This reduces our anticipated (14 × 14) system to (12 × 12), eliminating the displacement degrees of freedom at the attachment point and corner. Translating Newton's Law into rotational form and adding the rigid-elastic coupling method results in

$$I_{zz} \ddot{\theta} + \sum_{i=1}^{n} D_i \ddot{q}_i + I_w \ddot{\theta}_w = \tau_D \tag{24}$$

where:

$I_{zz}$ = body principal moment of inertia with respect to Z-axis;
$\ddot{\theta}$ = angular acceleration of the system rotation angle, $\theta$;
D = rigid-elastic coupling term;
$\ddot{q}$ = acceleration in generalized displacement coordinates;
$I_w$ = reaction wheel principal moment of inertia with respect to C, Z axis;
$\ddot{\theta}_w$ = angular acceleration of the reaction wheel rotation angle, $\theta_w$;
$T_D$ = disturbance torques.

Torques may be combined to resemble the basic expression of Newton's Law more closely:

$$I_{zz} \ddot{\theta} + \sum_{i=1}^{n} D_i \ddot{q}_i = \sum \tau \tag{25}$$

where $\sum \tau$ = the sum of the disturbance torques, and the control torques of the reaction wheel.

Isolating the first term:

$$\ddot{\theta} + \frac{\sum_{i=1}^{n} D_i}{I_{zz}} \ddot{q}_i = \frac{\sum \tau}{I_{zz}} \tag{26}$$

and also note:

$$\ddot{\theta} = \frac{\sum \tau}{I_{zz}} - \frac{\sum_{i=1}^{n} D_i}{I_{zz}} \ddot{q}_i \tag{27}$$

to aid repeatability. The natural frequencies and mode shape are nominal values (assuming accurate modeling). Time-varying natural frequencies driven by sudden changes

in system mass remain a challenge, which is the motivation for the techniques proposed in this article.

Time-Varying Natural Frequencies Due to Robot Grasping

A unique complication of applying classical methods to robotics is the inherent necessity of grasping and releasing objects (typically of unknowable mass), modifying the rigid body mode potentially nullifying, or at least reducing, the efficacy of the structural filters (to be described next). *This facet of space robotics strongly motivates limiting the frequency content of the control command (as proposed in this article) to avoid excitation of unknown new vibration modes*. The time-varying mass also motivates investigation utilization of the recently published deterministic artificial intelligence that accounts for both time-varying robotic system mass and frequency-dependent input shaping to avoid the deleterious effects of robot flexibility.

*2.3. Frequency Responses of Step and Periodic Functions*

The Fourier transform of the unit step function $f(t)$ is displayed in Equation (28), [24]. Notice the impulse occurrence is summed with an extra term with variable $\omega$ resulting in the response containing *all* the frequencies. This complication implies utilization of step functions as input trajectories, mandating subsequent actions to deal with the resonant response of exciting structural frequencies.

$$f(t) = \begin{cases} 1 & t \leq 0 \\ 0 & t > 0 \end{cases} \quad \leftrightarrow \quad F(\omega) = \int_{-\infty}^{0} e^{-j\omega t}dt = \int_{0}^{\infty} e^{j\omega t}dt = \pi\delta(\omega) - \frac{1}{j\omega} \tag{28}$$

*If you do a Fourier transform on the (time domain) function, which is zero for all time before t = 0 and 1 for all time after t = 0, the result is an infinite, continuous spectrum of frequency components. The power of each component is, of course, infinitesimally low, so you cannot measure anything. It is a very idealized function, and its (infinite) energy is spread over an infinite number of frequency components of infinitesimally small value* [29].

Meanwhile, note the Fourier transform of the sine function in Figure 9a compared to the frequency response function of the five-node space robot in Figure 9b. The challenge is much easier to surmount: to choose the fundamental frequency of the sine function to shape the input avoids the natural frequencies identified for the flexible space robot where Figure 10 illustrates harmonics to consider selecting the natural frequency.

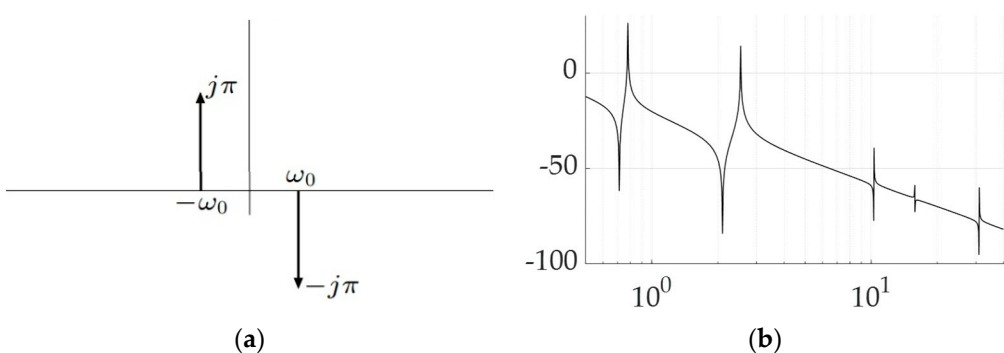

|  |  |
|:---:|:---:|
| (**a**) | (**b**) |

**Figure 9.** (**a**) Fourier transform of sine with frequency on the abscissa and response amplitude on the ordinate (**b**) sample robot frequency, and the response with frequency on the abscissa and response magnitude on the ordinate.

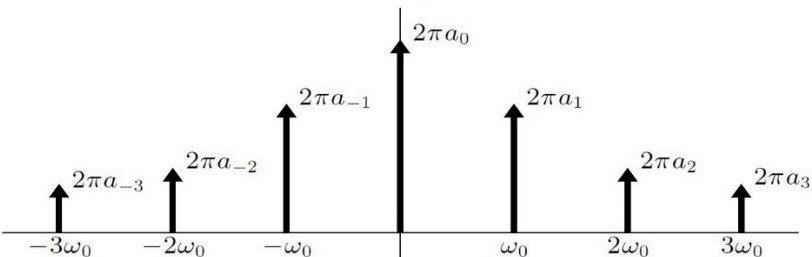

**Figure 10.** Harmonics used to properly place $\omega_0$. Fourier transform of the periodic signal with fundamental frequency $\omega_0$ with frequency on the abscissa, and the response amplitude on the ordinate. Notice the relatively easier (than using step functions) task of placing $\omega_0$ in (a) at locations other than the locations of the resonant and anti-resonant peaks of Figure 9. The task using harmonic signals is slightly more challenging due to the necessity to properly place $\omega_0$ and all the harmonics depicted in Figure 10.

### 2.4. Classical Control Approaches

Rather than use of piezo electric actuators along the robot structure, classical control of the end effector is achieved by controlling the central spacecraft's rigid body hub. Presented in the following sections are several options for achieving such control: classical gain stabilization, classical structural filtering, modern time-delay input shaping, recently proposed whiplash compensation, presently proposed rigid-body optimal input shaping, and alternative proposed frequency shaping of inputs.

*The implementation of the proposed frequency shaping of the inputs motivates the proposal of future research in the application of deterministic artificial intelligence, since the method is currently instantiated using the same frequency shaped inputs, and utilization of the method bestows the additional benefit of tracking time-varying natural frequencies (resonant and anti-resonant).*

### 2.5. Gain Stabilextion

Gain stabilization begins by asserting a specified form of the feedback control. Both classical and modern (state space and optimal) control instantiations ubiquitously impose $u = -Kx$, where the control is formulated as a negative feedback of state errors multiplied by user-designed gains. Controllers with proportional plus integral plus derivative natures are called PID controllers, and they remain quite popular due to their robustness, simple structure, ease of implementation, and continuing active research [39–42] in tuning the three gains associated with proportional, integral, and derivative components, respectively [43]. Equation (29) illustrates the equations for a control with proportional plus velocity components whose resulting closed-loop equation is displayed in Equation (30), which may be expressed in terms of active damping, active stiffness, and feedforward in Equation (31). Expression of the closed-loop system in the Laplace domain as a transfer function is represented in Equation (32).

$$u = K_p(\theta_d - \theta) + K_v\omega \ \rightarrow \ J\dot{\omega} = T_W \equiv u(t) \tag{29}$$

$$I\ddot{\theta} + K_V\dot{\theta} + K_P\theta = K_P\theta_d \tag{30}$$

$$I\ddot{\theta} + \underbrace{K_V\dot{\theta}}_{\substack{active \\ damping}} + \underbrace{K_P\theta}_{\substack{active \\ stiffness}} = \underbrace{K_P\theta_d}_{\substack{feed \\ forward}} \ \rightarrow \ \left[Is^2 + K_vs + K_p\right]\theta(s) = K_P\theta_d(s) \tag{31}$$

$$\frac{\theta(s)}{\theta_d(s)} = \frac{K_p}{Is^2 + K_vs + K_p} \ \rightarrow \ C.E.: \ s^2 + K_vs + K_p\Big|_{I=1} = s^2 + 2\xi\omega_n s + \omega_n{}^2 \tag{32}$$

Gains may be designed foremost for stabilization, and additionally for performance specifications, e.g., rise time, peak overshoot, settling time, etc. While these specifications

are met, residual oscillatory vibrations remain complicating during on-orbit robotic operations. The transfer function expression particularly reflects the natural frequencies of the system, and these frequencies guide the subsequent steps in the classical approach: structural filtering to tailor the compensation at these frequencies.

### 2.6. Classical Second-Order Structural Filtering

The compensation of the system elaborated in Equation (32) is often augmented by structural filters whose transfer function is represented by Equation (33), where the frequencies of the zeros and poles ($\omega_z$ and $\omega_p$, respectively) and damping of the zeros and poles ($\zeta_z$ and $\zeta_p$ respectively) are designable, leading to the steady state gain, phase, and maximum gain, as revealed in Equations (34)–(36). Figure 11 illustrates how notch and bandpass filters, respectively, may be implemented by choice of the four designable parameters: $\omega_z$, $\omega_p$, $\zeta_z$, and $\zeta_p$, where the maximum phase lead and lag occur at respective frequencies found in Equations (37) and (38).

$$\frac{Output(s)}{Input(s)} = \frac{\frac{s^2}{\omega_z^2} + \frac{2\zeta_z}{\omega_z^2}s + 1}{\frac{s^2}{\omega_p^2} + \frac{2\zeta_p}{\omega_p^2}s + 1} \tag{33}$$

$$K_\infty = 40log_{10}\left(\omega_p/\omega_z\right) \tag{34}$$

$$\phi_{max} = cos^{-1}\left[\frac{\left(2\zeta_c\sqrt{\omega_p/\omega_z}\right)^2 - \left(\omega_p/\omega_z - 1\right)^2}{\left(2\zeta_c\sqrt{\omega_p/\omega_z}\right)^2 + \left(\omega_p/\omega_z - 1\right)^2}\right] \tag{35}$$

$$K_{max} = 20log_{10}\left(\zeta_z/\zeta_p\right) dB \tag{36}$$

$$\omega_1/\omega_c = \sqrt{2\zeta_z\zeta_p + 1 - \sqrt{\left(2\zeta_z\zeta_p + 1\right)^2 - 1}} \tag{37}$$

$$\omega_2/\omega_c = \sqrt{2\zeta_z\zeta_p + 1 + \sqrt{\left(2\zeta_z\zeta_p + 1\right)^2 - 1}} \tag{38}$$

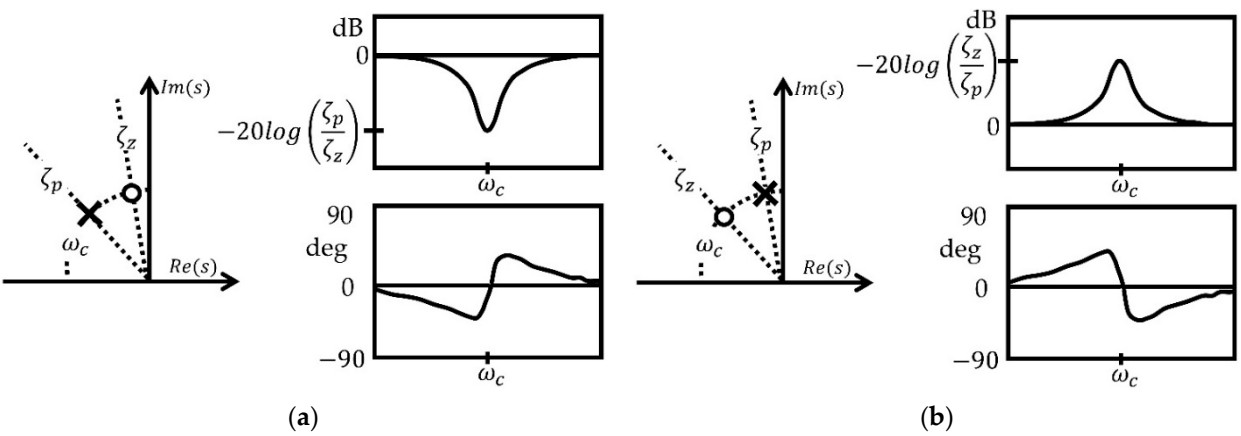

**(a)**　　　　　　　　　　　　　　**(b)**

**Figure 11.** Depictions of second-order structural filters. Graphics created by the author duplicating the depiction in [10]. (**a**) Notch filter named after the nature to "notch" out the response surrounding specific frequencies (e.g., $\omega_C$). (**b**) Bandpass filter named after the nature to only the response surrounding specific frequencies (e.g., $\omega_C$ ). Filters are placed in series, as depicted in Figure 12.

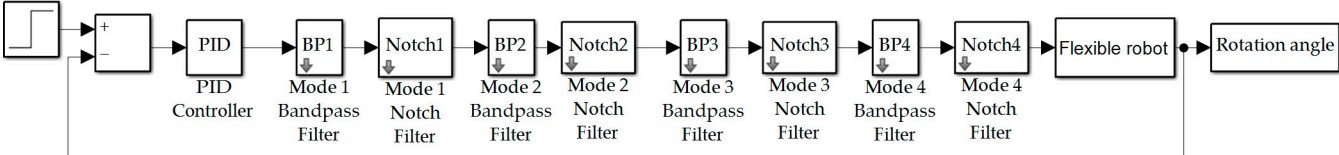

**Figure 12.** Topology of classical methods combining proportional, integral, derivative control with second-order structural filters (bandpass, abbreviated BP and notch) in accordance with Equations (33)–(36) and the depictions in Figure 13. Equations (25)–(27) may be used as the governing equations of motion for the flexible robot in the righthand side of Figure 12.

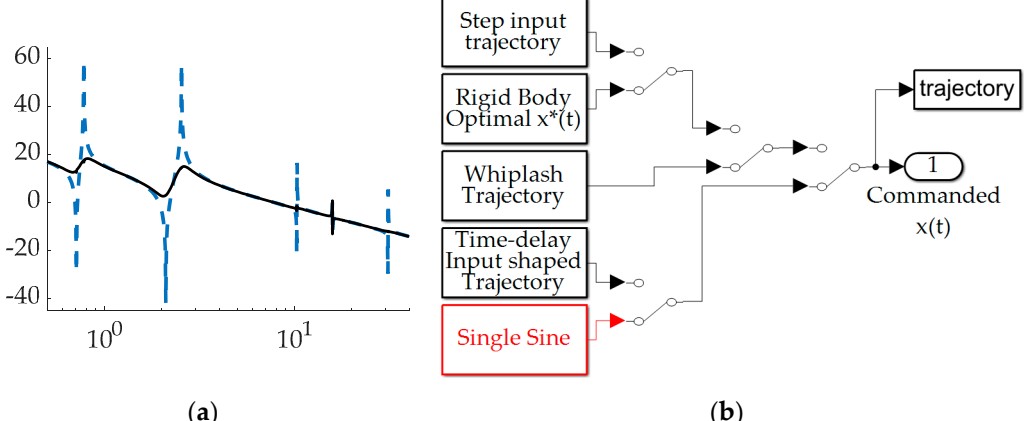

(**a**)            (**b**)

**Figure 13.** Compensation of multiple flexible modes of vibration (**a**), plotting the frequency response function, with frequency (in radians per second) on the abscissa and response in decibels on the ordinate. The dashed blue line is the original frequency response plot, while the solid black "flattened curve" is the compensated version. The context of this article's title is evident in this plot; (**b**) simulations created in SIMULINK to optionally switch between disparate methods facilitating direct comparison.

Incremental Application: Ensure Stability vs. Flatten the Curve

Subsequent application of the filters expressed by Equation (33) is displayed in Figure 12's depiction of the simulation blocks used to produce the results in Section 3. A debatable design question is which filters are recommended for implementation. Utilization of both notch and bandpass filters at each identified node would seek to flatten the curve (the amplitude plot of the frequency response).

Additional measures were investigated to control a presumably very highly flexible, lightweight space robot inspired by modern, optimal open-loop techniques, such as time-delay input shaping. A key strength is the avoidance of using step commands as the input trajectories, so several methods were compared, replacing the step input with the four alternatives depicted in Figure 13b: modern time-delayed input-shaping, rigid body optimal input trajectory, recently proposed whiplash compensation, and the presently proposed single sinusoidal input trajectory, where the single sine is chosen for both speed of performance and avoidance of flexible resonance and anti-resonance, as depicted in Figure 9b and the dashed line in Figure 13a. Each technique is described in the immediately following sections.

### 2.7. Whiplash Compensation Input Trajectory Shaping

So-called whiplash compensation was recently proposed [23] as the solution to the constrained optimization problem of controlling highly flexible robotics with initial control in the opposite direction than the desired end state, subsequently implementing a ramp function held until the desired final time. The system studied in [23] is identical to that studied here, displayed in Figure 8.

### 2.8. Rigid Body Optimal Input Trajectory Shaping

Trajectory optimization utilizing the methods of Pontryagin [39] is a competing option stemming from the just published [40], including nonlinear constrained optimization, real-time optimal control, and instantiations of such with and without singular switching and nonlinear decoupling of transport theorem terms. The intention in this article is to establish a new benchmark for comparison of these just published methods in a proposed sequel.

### 2.9. Single-Frequency Trajectory Shaping

So-called versine torque profiles [42] use smooth transcendental functions as a basis for "shaping" torque generation (such as the time-delayed input control shaping). With that inspiration, single-frequency sinusoids are proposed to instead shape autonomous trajectories by mimicking discontinuous step functions for flexible space robotics, as depicted in Figure 14. Two methods are displayed: (1) single-frequency implementation of whiplash trajectories, and (2) single-frequency shaping of step-like trajectories. Single-frequency sinusoids are also the current state-of-the-art trajectory shaper in deterministic artificial intelligence [24], making successful implementation in this article foreshadow a likely sequel research effort to integrate optimized control of flexible space robotics with deterministic artificial intelligence.

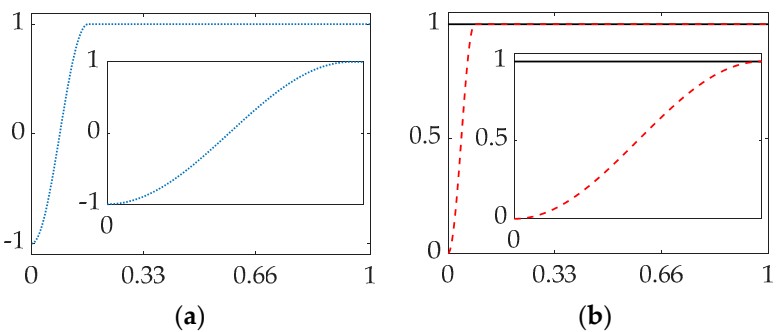

**(a)**　　　　　　**(b)**

**Figure 14.** Alternative frequency trajectory shaping proposals: (**a**) nonlinear, optimal whiplash trajectory shaping; (**b**) single-frequency trajectory shaping. Notice the unexpected opposite initial direction of the commanded optimal "whiplash" trajectory.

### 2.10. Deterministic Artificial Intelligence

#### 2.10.1. Self-Awareness Statements

The foremost premise of deterministic artificial intelligence is to assert a physics-based [26] statement of self-awareness [27] comprised of the governing differential equations of motion (e.g., Equations (35)–(38)) expressed in terms of the desired trajectories [24], which must be analytic expressions to facilitate autonomous calculation. The desired trajectories are currently formulated using single sinusoidal trajectories [24], as described in Sections 2.7–2.9, while the current novel developments seek to utilize constrained optimal trajectories [44] in hopes of favorable comparison to sinusoidal trajectories.

$$F = m\ddot{x} + m\frac{d\omega}{dt} \times r\prime + 2m\omega \times v\prime + m\omega \times (\omega \times r\prime) \tag{39}$$

$$u \equiv m\ddot{x}_d + m\frac{d\omega_d}{dt} \times r\prime_d + 2m\omega_d \times v\prime_d + m\omega_d \times (\omega_d \times r\prime_d) \tag{40}$$

$$-T_W = J\dot{\omega} + \omega \times J\omega \rightarrow u \equiv J\dot{\omega}_d + \omega_d \times J\omega_d \tag{41}$$

$$I_{zz}\ddot{\theta} + \sum_{i=1}^{n} D_i\ddot{q}_i + I_w\ddot{\theta}_w = T_D \rightarrow u \equiv I_{zz}\ddot{\theta}_d + \sum_{i=1}^{n} D_i\ddot{q}_{i_d} + I_w\ddot{\theta}_{w_d} \tag{42}$$

The modification of the governing differential equations of motion by replacing the motion states with desired states necessitates an autonomous desired state trajectory

generation, and this partially motivates the application of single-sinusoidal frequency-based state trajectories. Development of such trajectories in this article make possible future sequel research applying deterministic artificial intelligence (as currently instantiated) to the optimal control of highly flexible space robotics.

### 2.10.2. Adaption or Optimal Learning

By reparametrizing Equations (35)–(38) into regression form, optimal learning [27] or simple adaption [26] may be used to update the regression vector of unknowns (typically including variables such as mass, mass moments of inertia, and environmental properties, such as those of the atmosphere, magnetic field, gravitational acceleration, and solar radiation).

### 2.11. Simulation Parameters

MATLAB/SIMULINK release 2021b was used to create the simulations included in the Appendix, whose results are presented in Section 3. The Runge–Kutta integration solver was used with a fixed time-step of 0.01 s.

### 3. Results

This section provides a concise and precise description of the experimental results, their interpretation, as well as the experimental conclusions that can be drawn. Section 3.1 establishes a classical benchmark: gain stabilization using proportional, integral, derivative control. Section 3.2 augments Section 3.1 by adding second-order classical structural filters (notch and bandpass sequentially). Filters are added incrementally in Sections 3.2–3.5 for each structural mode from low frequency to high frequency, *seeking to ascertain the minimal necessary instantiation*, culminating in a comparisons of stability margins in Section 3.6. Results using trajectory shaping are presented in Section 3.7 followed by direct comparison in Section 4.

### 3.1. Gain Stabilization Using Proportional Plus Integral Plus Derivative Control

Gain stabilization is the established performance benchmark, and the results will appear in subsequent comparisons, incrementally adding additional compensation. Results are qualitatively compared to the benchmark in step response plots and frequency response plots with quantitative comparisons using gain margin and phase margin for stability and mean tracking errors for performance.

### 3.2. Gain Stabilization Plus Structural Filters (Bandpass and Notch) for the First Flexible Mode

Gain stabilization was augmented with structural filters using Equation (33), where structural resonant frequencies were used as filter natural frequencies and filter damping was adjusted to visually "flatten the curve" of the frequency response plot. First, a bandpass variant was designed for the first anti-resonant spike. Next, a notch variant was designed for the first resonant spike. Filter details are available in the Appendix B to aid repeatability. Figure 15 displays the results. Notice in the step response plot in Figure 15, the nominal performances are nearly indistinguishable, while in the "inset" plot distinct differences are apparent, where quantitative results are displayed in Table 1. Figure 16 displays the four frequency response plots overlaid. The nominal frequency response (prior to gain stabilization) is lowest, indicated by the dashed blue line. The thin-solid red line indicates a gain-stabilized case. Meanwhile, the overlaid dashed gold line indicates bandpass compensation of the first anti-resonance. Finally, the "flattened" frequency response plot's thin, dotted black line indicates full mode compensation with the addition of a notch filter at the first structural resonant frequency. Effects on the frequency response phase is displayed in Table 1. Addition of the bandpass filter at the first anti-resonance leads to the highest gain margin, with an excellent phase margin, while addition of the notch filter at the first resonance increases the phase margin substantially further with a slight decrease in gain stability.

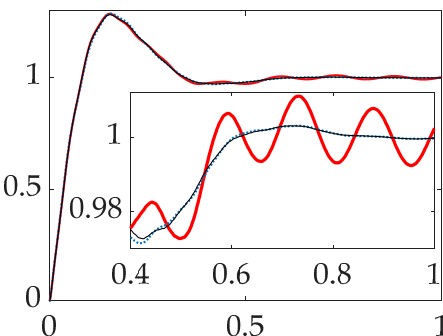

**Figure 15.** Five computer simulations of the control of rigid body; the compensated first flexible anti-resonance, compensated first resonance, compensated second anti-resonance, and fully compensated first two modes, with frequency in hertz on the abscissa and response magnitude in degrees on the ordinate. PID (thick, solid red line); PID + bandpass 1 (dotted blue line); and PID + bandpass 1 + notch 1 (thin, solid black line) illustrating substantial improvement by compensating for first anti-resonance, but negligible difference also compensating for first resonance.

**Table 1.** Compensation of rigid body plus first flexible mode (resonance and anti-resonance) with step input.

| System | Gain Margin (dB) | Phase Margin (Degrees) | Stable/Unstable |
|---|---|---|---|
| Uncontrolled | 64.5 | 0.000673 | Stable |
| PID controlled | 4.95 | 17.9 | Stable |
| PID + bandpass | −26.2 [1] | 80.2 | Stable |
| PID + bandpass + notch | −25.9 [1] | 133 | Stable |

[1] Negative gain margins indicate that stability is lost by decreasing the gain, while positive gain margins indicate that stability is lost by increasing the gain.

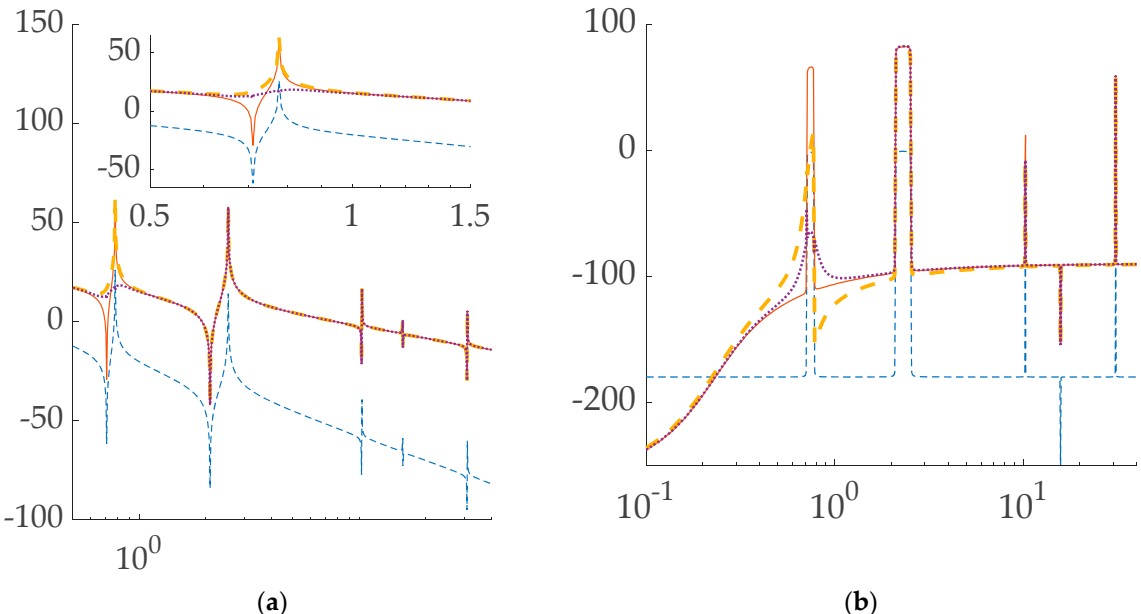

(**a**)  (**b**)

**Figure 16.** Four computer simulations of the control of rigid body and first flexible mode with frequency in hertz on the abscissa: uncontrolled (narrow, dashed line), PID compensated (thin, solid line), PID compensated with bandpass filter of first antiresonance (thick, dashed line), and PID compensated with bandpass and notch filter of first mode—anti-resonance and resonance (dotted line). (**a**) Response magnitude (first mode zoomed in the inset) on the ordinate; (**b**) phase in degrees on the ordinate.

### 3.3. Gain Stabilization Plus Structural Filters (Bandpass and Notch) for Two Flexible Modes

Gain stabilization with the fully compensated first mode was augmented with structural filters using Equation (33), where the structural resonant frequencies were used as filter natural frequencies and filter damping was adjusted to visually "flatten the curve" of the frequency response plot at the location of the second mode. First, a bandpass variant was designed for the second anti-resonant spike. Next, a notch variant was designed for the second resonant spike. Figure 17 displays the results. Notice once again in the step response plot in Figure 17 the nominal performances are nearly indistinguishable, while in the "inset" plot distinct differences are apparent, where quantitative results are displayed in Table 2. Figure 18 displays the four frequency response plots overlaid. The nominal frequency response (after gain stabilization) is indicated by the dashed blue line. The thin, solid red line indicates the addition of full compensation of the first resonant mode. Meanwhile, the overlaid dashed gold line indicates bandpass compensation of the second anti-resonance. Finally, the "flattened" frequency response plot's thin, dotted black line indicates full mode compensation with the addition of a notch filter at the first two structural resonant frequencies. Effects on frequency response phase is displayed in Table 2. Addition of both bandpass filter at the second anti-resonance and notch filter at the second resonance increases reduces both the gain and phase margin substantially. Compensation of the second resonant structure mode should therefore be avoided, despite an instinct to "flatten the curve" of the frequency response plot magnitude.

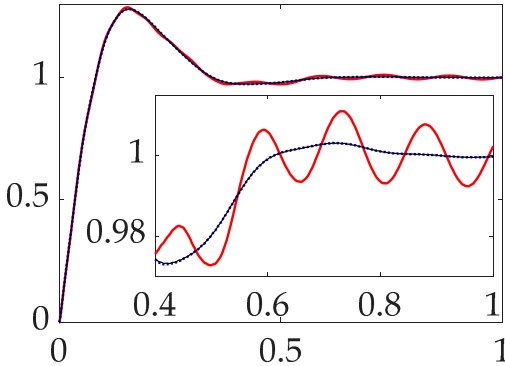

**Figure 17.** Five computer simulations of the control of rigid body; the compensated first flexible mode, compensated second anti-resonance, and fully compensated first two modes, with frequency in hertz on the abscissa and response magnitude in degrees on the ordinate. PID (thick, solid red line); PID + mode 1 + bandpass 2 (dotted blue line); PID + mode 1 + bandpass 2 + notch 2 (thin, solid black line), again illustrating substantial improvement by compensating for additional (second) anti-resonance, but negligible difference also compensating for the first additional (second). Table 2 quantitative data corresponding to qualitative presentation in Figure 17.

**Table 2.** Compensation of rigid body plus first two flexible modes (resonance and anti-resonance) with step input.

| System | Gain Margin (dB) | Phase Margin (Degrees) | Stable/Unstable |
|:---:|:---:|:---:|:---:|
| Uncontrolled | 64.5 | 0.000673 | Stable |
| PID controlled | 4.95 | 17.9 | Stable |
| **PID + mode 1** | **−25.9** [1] | **133** | **Stable** |
| PID + mode 1 + mode 2 | 5.52 | 18.3 | Stable |
| PID + mode 1 + mode 2 + bandpass | 0.625 | 4.01 | Stable |

[1] Negative gain margins indicate that stability is lost by decreasing the gain, while positive gain margins indicate that stability is lost by increasing the gain.

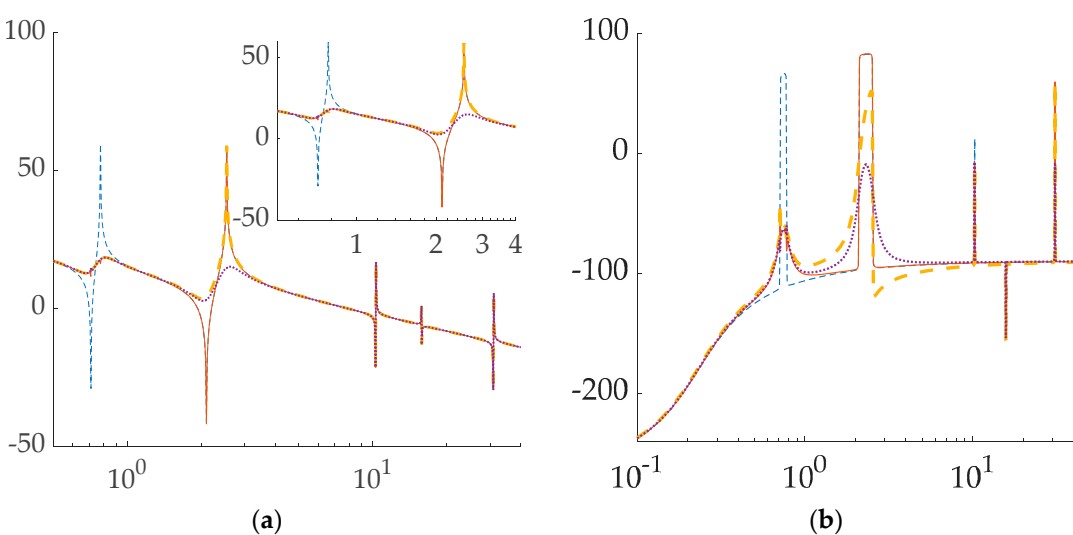

**Figure 18.** Four computer simulations of control of the rigid body; first flexible mode and second flexible mode, with frequency in hertz on the abscissa: uncontrolled (narrow, dashed line); PID compensated, filtered first mode (thin, solid line); PID compensated, filtered first mode, plus bandpass filter of second antiresonance (thick, dashed line); and PID compensated, filtered first mode and second mode—anti-resonance and resonance (dotted line). (**a**) Response magnitude (first mode zoomed in the inset) on the ordinate; (**b**) phase in degrees on the ordinate.

### 3.4. Gain Stabilization Plus Structural Filters (Bandpass and Notch) for the First Three Flexible Modes

Gain stabilization with the fully compensated first mode was augmented with structural filters using Equation (33), where the structural resonant frequencies were used as filter natural frequencies and filter damping was adjusted to visually "flatten the curve" of the frequency response plot at the location of the second mode. First, a bandpass variant was designed for the second anti-resonant spike. Next, a notch variant was designed for the second resonant spike. Figure 19 displays the results. Notice once again in the step response plot in Figure 19 that the nominal performances are nearly indistinguishable, while in the "inset" plot distinct differences are apparent, where quantitative results are displayed in Table 3. Figure 20 displays the four frequency response plots overlaid. The nominal frequency response (after gain stabilization) is indicated by the dashed blue line. The thin, solid red line indicates addition of full compensation of the first resonant mode. Meanwhile, the overlaid dashed gold line indicates bandpass compensation of the second anti-resonance. Finally, the "flattened" frequency response plot's thin, dotted black line indicates full mode compensation, with the addition of a notch filter at the first three structural resonant frequencies. Effects on the frequency response phase is displayed in Table 3. As with the case of compensating for the second structural modes, addition of both a bandpass filter at the third anti-resonance and notch filter at the third resonance increases both the gain and phase margins substantially. Compensation of the third resonant structure mode should therefore be avoided, despite an instinct to "flatten the curve" of the frequency response plot magnitude.

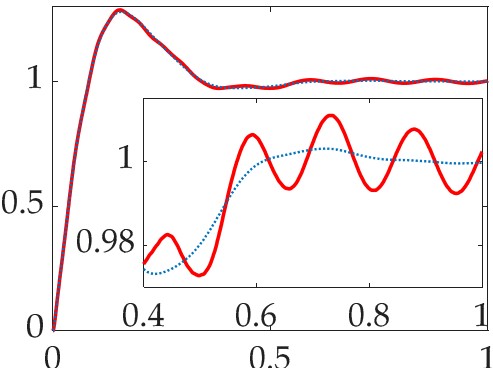

**Figure 19.** Five computer simulations of control of the rigid body; compensated first flexible mode; compensated second anti-resonance; and fully compensated first two modes with frequency in hertz on the abscissa and response magnitude in degrees on the ordinate. Table 3 quantitative data corresponding to qualitative presentation in Figure 19. PID (thick, solid red line); PID + modes 1–2 + bandpass 3 (dotted blue line); PID + modes 1–3 (thin, solid black line) again illustrating substantial improvement by compensating for additional (second) anti-resonance, but negligible difference also compensating for first additional (second).

**Table 3.** Compensation of rigid body plus first three flexible modes (resonance and anti-resonance) with step input.

| System | Gain Margin (dB) | Phase Margin (Degrees) | Stable/Unstable |
|---|---|---|---|
| Uncontrolled | 64.5 | 0.000673 | Stable |
| PID controlled | 4.95 | 17.9 | Stable |
| **PID + mode 1** | **−25.9** [1] | **133** | **Stable** |
| PID + mode 2 + bandpass | 3.87 | 16.4 | Stable |
| PID + mode 2 + mode 3 | 5.52 | 18.3 | Stable |

[1] Negative gain margins indicate that stability is lost by decreasing the gain, while positive gain margins indicate that stability is lost by increasing the gain.

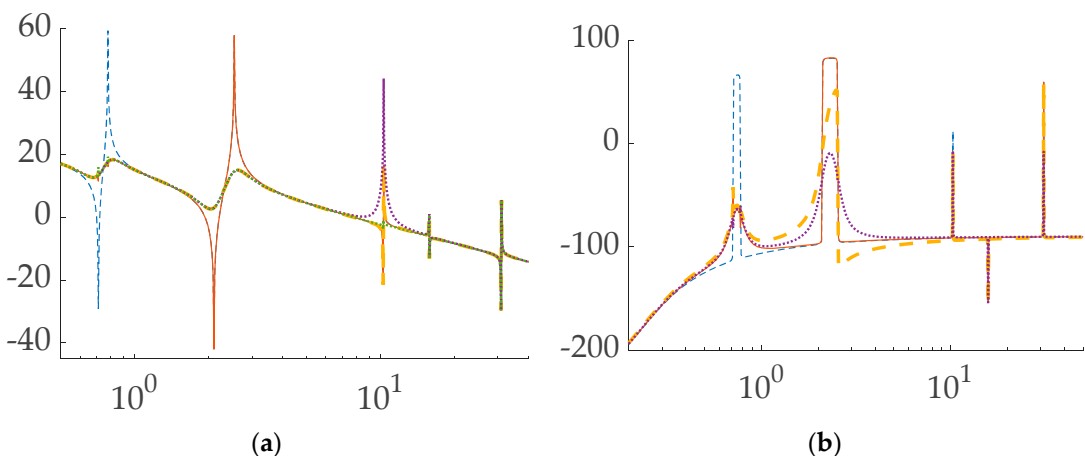

(**a**)　　　　　　　　　　　(**b**)

**Figure 20.** Six computer simulations of control of the rigid body, first flexible mode, second flexible mode, and third flexible mode with frequency in hertz on the abscissa: uncontrolled (narrow, dashed line); PID compensated, filtered first mode (thin, solid line); PID compensated, filtered first mode; PID compensated, filtered first and second mode (thick, dashed line); and PID compensated, filtered first, second, and third mode—anti-resonance and resonance (dotted line), with frequency in radians per second on the abscissa. (**a**) Response magnitude (first mode zoomed in the inset) on the ordinate; (**b**) phase in degrees on the ordinate.

### 3.5. Gain Stabilization Plus Structural Filters (Bandpass and Notch) for the First Four Flexible Modes

Gain stabilization with the fully compensated first mode was augmented with structural filters using Equation (33), where structural resonant frequencies were used as filter natural frequencies and filter damping was adjusted to visually "flatten the curve" of the frequency response plot at the location of the second mode. First, a bandpass variant was designed for the second anti-resonant spike. Next, a notch variant was designed for the second resonant spike. Figure 21 displays the results. Notice once again in the step response plot in Figure 21 that the nominal performances are nearly indistinguishable, while in the "inset" plot distinct differences are apparent, where the quantitative results are displayed in Table 4. Figure 22 displays the four frequency response plots overlaid. The nominal frequency response (after gain stabilization) is indicated by the dashed blue line. The thin, solid red line indicates addition of full compensation of the first resonant mode. Meanwhile, the overlaid dashed gold line indicates bandpass compensation of the second anti-resonance. Finally, the "flattened" frequency response plot's thin, dotted black line indicates full mode compensation with the addition of a notch filter at the first three structural resonant frequencies. Effects on frequency response phase is displayed in Table 4. Different than the cases of compensating for the second and third structural modes, addition of the bandpass filter at the fourth anti-resonance restores the outstanding gain margin performance, but addition of the notch filter at the fourth resonance does not restore the phase margin performance. Compensation of the fourth resonant structure mode does not exceed the nominal performance, compensating only for the first vibrational mode and should therefore be avoided, despite an instinct to "flatten the curve" of the frequency response plot magnitude.

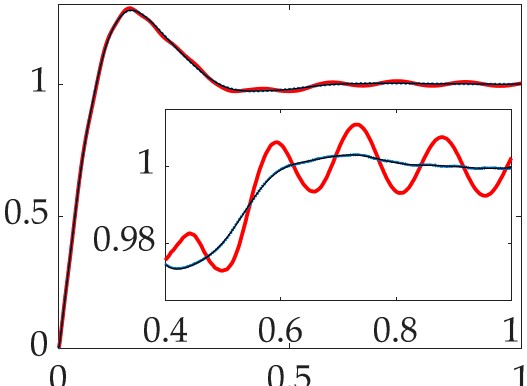

**Figure 21.** Five computer simulations of control of the rigid body and fully compensated modes with frequency in hertz on the abscissa and response magnitude in degrees on the ordinate. Table 4 quantitative data corresponding to qualitative presentation in Figure 21. PID (thick, solid red line); PID + modes 1–3 + bandpass 4 (dotted blue line); PID + modes 1–4 (thin, solid black line) again illustrating substantial improvement by compensating for additional (second) anti-resonance, but negligible difference also compensating for first additional (second).

### 3.6. Comparisons of Stability Margins: Gain Stabilization Plus Structural Filters (Bandpass and Notch)

This section summarizes the incrementally achieved results thus far in Table 5. Noteworthy performances are highlighted in bold font, indicating compensation of the first mode alone following gain compensation is the superior method based on a combined measure of both the gain and phase margin. Meanwhile, additional compensation of the first three structural modes plus the fourth anti-resonance produce the highest gain margin but suffer from a reduction in the phase margin.

**Table 4.** Compensation of rigid body plus four flexible modes (resonance and anti-resonance) with step input.

| System | Gain Margin (dB) | Phase Margin (Degrees) | Stable/Unstable |
| --- | --- | --- | --- |
| Uncontrolled | 64.5 | 0.000673 | Stable |
| PID controlled | 4.95 | 17.9 | Stable |
| PID + mode 1 | −25.9 [1] | **133** | Stable |
| PID + modes 1–2 | 5.52 | 18.3 | Stable |
| PID + modes 1–3 | 5.89 | 18.4 | Stable |
| PID + modes 1–3 + bandpass 4 | **−26.8** | 17 | Stable |
| PID + modes 1–4 | 6.1 | 18.5 | Stable |

[1] Negative gain margins indicate that stability is lost by decreasing the gain, while positive gain margins indicate that stability is lost by increasing the gain.

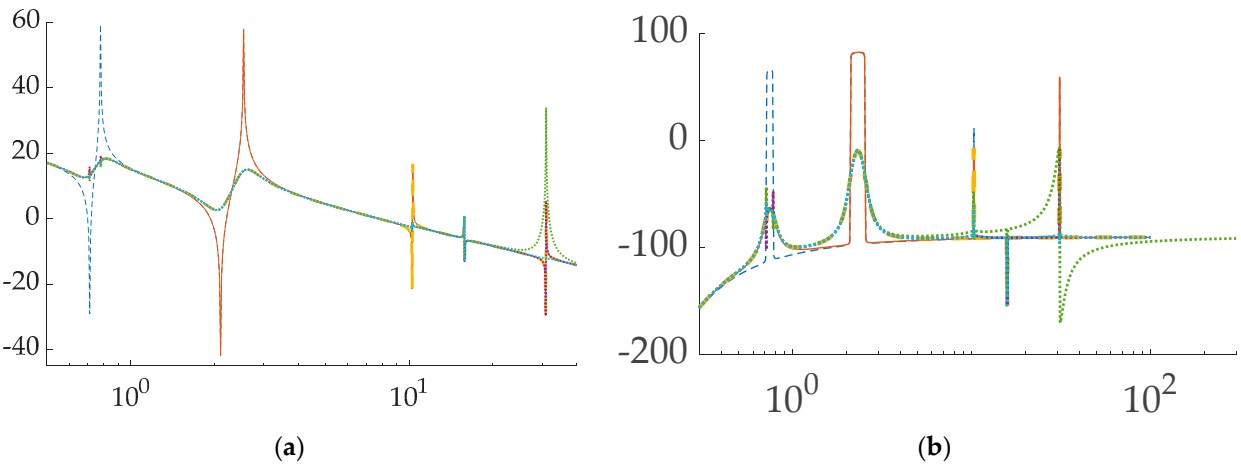

(**a**)        (**b**)

**Figure 22.** Six computer simulations of control of the rigid body; first flexible mode, second flexible mode, third flexible mode, and fourth flexible mode with frequency in hertz on the abscissa: uncontrolled (narrow, dashed line); PID compensated, filtered first mode (thin, solid line); PID compensated, filtered first mode; PID compensated, filtered first and second mode (thick, dashed line); and PID compensated, filtered first, second, and third mode—anti-resonance and resonance (dotted line). (**a**) Response magnitude (first mode zoomed in the inset) on the ordinate; (**b**) phase in degrees on the ordinate.

**Table 5.** Classical compensation of the rigid body plus first four flexible modes (resonances and anti-resonances), with the step inputs.

| System | Percent Improved Gain Margin [1] | Percent Improved Phase Margin |
| --- | --- | --- |
| PID controlled | - - | - - |
| PID + mode 1 | **423.4%** | **643.0%** |
| PID + mode 1 + mode 2 | 11.5% | 2.2% |
| PID + mode 1 + mode 2 + mode 3 | 19.0% | 2.8% |
| PID + mode 1 + mode 2 + mode 3 + bandpass | **441.4%** | **−5.0%** |
| PID + mode 1 + mode 2 + mode 3 + mode 4 | 23.2% | 3.4% |

[1] Negative gain margins indicate that stability is lost by decreasing the gain, while positive gain margins indicate that stability is lost by increasing the gain.

From these results comes the practice of not fully compensating for all flexible modes in favor of compensating only for the first flexible mode in addition to benchmark gain

stabilization. Next, consider utilizing single sinusoidally shaped step inputs to improve the target-tracking performance.

### 3.7. Sinusoidal Trajectory Shaping: Gain Stabilization Plus Structural Filters of Only Selected Modes

Assuming the best performing compensator identified in Section 3.6 (benchmark gain stabilization with classical compensation of the first flexile mode's anti-resonance and resonance), this section addresses shaping the step command to further increase performance, where the results are depicted in Figure 23, comparing the step trajectories versus the single-sinusoidal shaped-trajectories mimicking the step functions. The figure of merits used for comparison include the gain margin and phase margin (listed in Table 5), and a trajectory tracking error in Table 6 with percentage performance improvement.

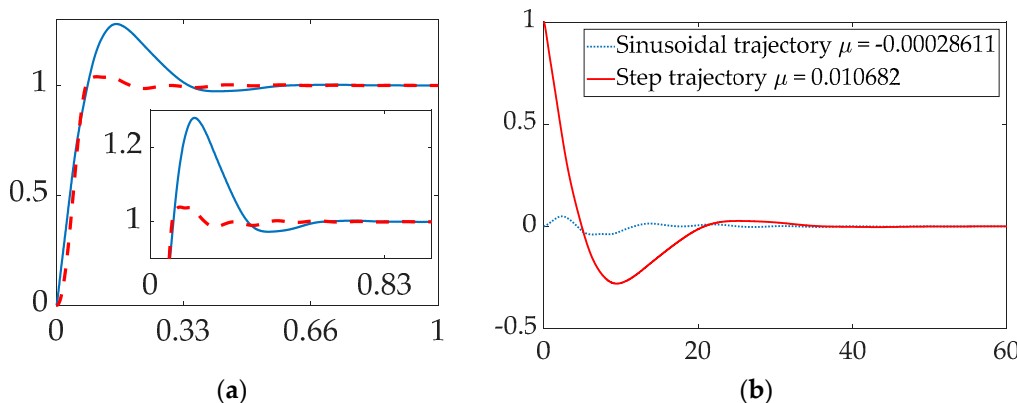

(a)          (b)

**Figure 23.** Utilizing sinusoidal-shaped trajectories to improve the step response: (**a**) responses for step and sinusoidal trajectories; and (**b**) tracking error following sinusoidal and step trajectories.

**Table 6.** Controlling rigid body with four flexible modes (resonance and anti-resonance) with step inputs.

| Shaping Method | Mean Tracking Error (Deviation) | Percent Performance Improvement |
|---|---|---|
| No trajectory shaping | 0.010682 | - - |
| **Single sine shaping** | **0.00028611** | **97** |
| Shaped whiplash | 0.014454 | 35 |
| Time-delayed shaping | 0.015681 | 47 |

Notice sinusoidal trajectory shaping is compared to two other modern benchmarks in Figure 24: shaped-whiplash and time-delay input shaped. The latter two methods do not improve the performance when combined with classical feedback compensation, which seems logical since the methods are derived to act alone. Meanwhile, trajectory shaping with single-frequency sinusoids with simultaneous classical mode compensation improve the performance two orders of magnitude.

Section 3 includes the simulation results presented in both qualitative fashion vis multi-plots, but also quantitatively using ubiquitous figures of merit: gain margin, phase margin, mean error, and finally the percent performance improvement. Conclusions include:

1. *Superior performance (based on gain margin, phase margin, and mean tracking error) is attained by only compensating for the first complete flexible mode (resonance and anti-resonance) with notch and bandpass filters, respectively, with step commands shaped by novel single-sinusoidal trajectories.*

   a. The next-best performer was the novel shaped-whiplash control based on [23].

2. Specific compensation of higher modes did not substantially improve the performance, with the exception of fully compensating the rigid body, the first three flexible modes

and bandpass filter, and fourth the anti-resonance. This option very substantially improved the gain margin, while slightly decreasing the phase margin.

The results from Section 3 are summarized in general terms in the discussion of Section 4, emphasizing the percent performance improvement over the declared benchmark.

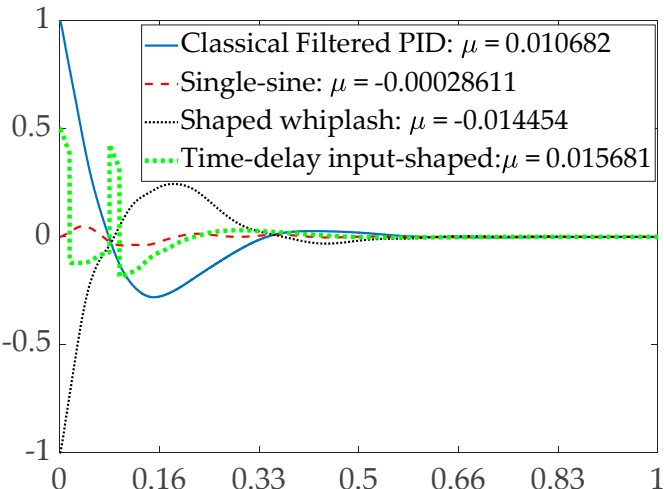

**Figure 24.** Comparing whiplash compensation, time-delay shaping, filtered PID (with step trajectories), and single-sinusoidal trajectories: error in degrees on the ordinate with displacement on the abscissa; Table 6 means tracking errors and performance improvement percentages.

## 4. Discussion

Instincts to improve system performance by addressing maladies are tempered by the curse of dimensionality. As more and more maladies are addressed, the system order (of the governing differential equations) increases and so does the complication in coupled systems of differential equations. This notion manifests in two regards: stability and trajectory-tracking performance. In terms of stability, the system is enhanced most by compensating for the first flexible mode alone, with deleterious effects resulting from additional compensation of higher vibrational modes. On the other hand, target tracking performance increases with system order. Augmentation of the nominal step inputs (Table 7) with sinusoidal shaped trajectories (Table 8) leads to substantially larger improvements than using step functions and furthermore achieves the best performance at a relatively lower system order.

**Table 7.** The mean trajectory-tracking error of the step responses.

| System with *Step Input* | System Order | Mean Error | Percent Improvement |
|---|---|---|---|
| PID controlled with step input (reference) | 26 | 0.013984 | - - |
| PID + mode 1 | 34 | 0.013848 | 0.97 |
| PID + mode 1 + mode 2 | 42 | 0.010795 | 22.8 |
| **PID + mode 1 + mode 2 + mode 3** | **50** | **0.010682** | **23.6** |

### 4.1. Comparison(s) Summary: Mean Errors and Percent Performance Improvements

Table 7 indicates that the classical mode compensation increases the performance over twenty percent, while addition of sinusoidally shaped trajectories raises the performance increase to well over ninety percent at lower system orders, as displayed in Table 8. The best results are achieved by compensating for the first complete flexible mode (both resonance and anti-resonance) with bandpass and notch filters, respectively, and shaping the

commanded trajectories with single-frequency sinusoids. An over ninety-eight percent performance improvement is achieved compared to classical PID control with step inputs.

**Table 8.** The mean tracking errors using the step responses with sinusoidal-shaped trajectories.

| System with *Single-Sinusoidal Shaped Trajectory* | System Order | Mean Error | Percent Improvement |
|---|---|---|---|
| PID controlled with step input (benchmark) | 26 | 0.013984 | - - |
| PID controlled | 26 | 0.000502 | 96.4 |
| **PID + mode 1** | **34** | **0.000199** | **98.6** |
| PID + mode 1 + mode 2 | 42 | 0.00021 | 98.4 |
| PID + mode 1 + mode 2 + mode 3 | 50 | −0.00029 | 98.0 |

> **RECOMMENDATION:** *Use single-frequency, sinusoidally shaped step functions as inputs to flexible space robots and compensate for flexible modes classically as preferred.*

### 4.2. Future Research Recommendations

Particularly in light of the superior performance of single-sine shaped input trajectories, deterministic artificial intelligence is a natural recommendation for sequel research, especially since the method's current instantiation emphasizes single-sine trajectory shaping. The combination of deterministic artificial intelligence will leverage the work presented in this article to track time-varying natural vibration modes as a matter of fact of the two methods' shared formulation.

Additionally, the findings indicate the system wants to be driven from the initial state to the terminal state by a function with minimum curvature, or with the minimal integral of squared acceleration. An intriguing notion is investigation of the sinusoidal, near-optimum approximation by cubic polynomials.

**Funding:** This research received no external funding. The APC was funded by the author.

**Data Availability Statement:** Data supporting reported results can be obtained by contacting the author.

**Acknowledgments:** Department of Defense and NASA content (images, videos, audio, etc.) are generally not copyrighted and may be used for educational or informational purposes without needing explicit permissions.

**Conflicts of Interest:** The author declares no conflict of interest.

### Appendix A. Elaboration of Modal System Identification on the Flexible Robot System

Substitute into Equation (19):

$$\ddot{\theta} + \frac{\sum_{i=1}^{n} D_i(-2\zeta\omega_i I_{zz})}{I_{zz}(I_{zz} - D_i\sum_{i=1}^{n} D_i)}\dot{q}_i + \frac{\sum_{i=1}^{n} D_i(\omega_i^2 I_{zz})}{I_{zz}(I_{zz} - D_i\sum_{i=1}^{n} D_i)}q_i + \frac{\sum_{i=1}^{n} D_i(-D_i T)}{I_{zz}(I_{zz} - D_i\sum_{i=1}^{n} D_i)} = \frac{T}{I_{zz}} \quad \text{(A1)}$$

$$\ddot{\theta} = \frac{2\zeta\omega_i\sum_{i=1}^{n} D_i}{I_{zz} - D_i\sum_{i=1}^{n} D_i}\dot{q}_i + \frac{\omega_i^2\sum_{i=1}^{n} D_i}{I_{zz} - D_i\sum_{i=1}^{n} D_i}q_i + \frac{TD_i\sum_{i=1}^{n} D_i}{I_{zz}(I_{zz} - D_i\sum_{i=1}^{n} D_i)} = \frac{T}{I_{zz}} \quad \text{(A2)}$$

$$\ddot{\theta} = \frac{2\zeta\omega_i\sum_{i=1}^{n} D_i}{I_{zz} - D_i\sum_{i=1}^{n} D_i}\dot{q}_i + \frac{\omega_i^2\sum_{i=1}^{n} D_i}{I_{zz} - D_i\sum_{i=1}^{n} D_i}q_i + \frac{T(D_i\sum_{i=1}^{n} D_i + I_{zz} - D_i\sum_{i=1}^{n} D_i)}{I_{zz}(I_{zz} - D_i\sum_{i=1}^{n} D_i)} \quad \text{(A3)}$$

$$\ddot{\theta} = \frac{2\zeta\omega_i\sum_{i=1}^{n} D_i}{I_{zz} - D_i\sum_{i=1}^{n} D_i}\dot{q}_i + \frac{\omega_i^2\sum_{i=1}^{n} D_i}{I_{zz} - D_i\sum_{i=1}^{n} D_i}q_i + \frac{TI_{zz}}{I_{zz}(I_{zz} - D_i\sum_{i=1}^{n} D_i)} \quad \text{(A4)}$$

$$\ddot{\theta} = \frac{2\zeta\omega_i \sum_{i=1}^{n} D_i}{I_{zz} - D_i \sum_{i=1}^{n} D_i}\dot{q}_i + \frac{\omega_i^2 \sum_{i=1}^{n} D_i}{I_{zz} - D_i \sum_{i=1}^{n} D_i}q_i + \frac{T}{\left(I_{zz} - D_i \sum_{i=1}^{n} D_i\right)} \tag{A5}$$

Recall the expressions for the rigid elastic coupling using modal coordinates: $D_i = \int_F \left(x_F \phi_i^y - y_F \phi_i^x\right) dm$, where $\phi$'s are the mode shapes from finite element analysis using the eigenvalues of K/m (stiffness/mass). The system stiffness matrix is included in Table A1 and the mass matrix in Table A2 shows the natural frequencies and mode shapes for the flexible system in Tables A3 and A4.

**Table A1.** Stiffness matrix (K). [1]

|  | W2 | θ2 | W3 | θ3 | W4 | θ4 | W5 | θ5 | U6 | θ6 | u7 | θ7 | u8 | θ8 |
|---|---|---|---|---|---|---|---|---|---|---|---|---|---|---|
| W2 | 958.8179 | 0.0000 | −479.409 | 59.9261 | 0 | 0 | 0 | 0 | 0 | 0 | 0 | 0 | 0 | 0 |
| θ2 | 0.0000 | 19.9754 | −59.9261 | 4.9938 | 0 | 0 | 0 | 0 | 0 | 0 | 0 | 0 | 0 | 0 |
| W3 | −479.409 | −59.926 | 958.8179 | 0.0000 | −479.409 | 59.9261 | 0 | 0 | 0 | 0 | 0 | 0 | 0 | 0 |
| θ3 | 59.9261 | 4.9938 | 0.0000 | 19.9754 | −59.9261 | 4.9938 | 0 | 0 | 0 | 0 | 0 | 0 | 0 | 0 |
| W4 | 0 | 0 | −479.409 | −59.926 | 958.8179 | 0.0000 | −479.409 | 59.9261 | 0 | 0 | 0 | 0 | 0 | 0 |
| θ4 | 0 | 0 | 59.9261 | 4.9938 | 0.0000 | 19.9754 | −59.9261 | 4.9938 | 0 | 0 | 0 | 0 | 0 | 0 |
| W5 | 0 | 0 | 0 | 0 | −479.409 | −59.926 | 479.409 | −59.926 | 0 | 0 | 0 | 0 | 0 | 0 |
| θ5 | 0 | 0 | 0 | 0 | 59.9261 | 4.9938 | −59.9261 | 19.9754 | −59.9261 | 4.9938 | 0 | 0 | 0 | 0 |
| U6 | 0 | 0 | 0 | 0 | 0 | 0 | 0 | −59.926 | 958.8179 | 0.0000 | −479.409 | 59.9261 | 0 | 0 |
| θ6 | 0 | 0 | 0 | 0 | 0 | 0 | 0 | 4.9938 | 0.0000 | 19.9754 | −59.9261 | 4.9938 | 0 | 0 |
| U7 | 0 | 0 | 0 | 0 | 0 | 0 | 0 | 0 | −479.409 | −59.926 | 958.8179 | 0.0000 | −479.409 | 59.9261 |
| θ7 | 0 | 0 | 0 | 0 | 0 | 0 | 0 | 0 | 59.9261 | 4.9938 | 0.0000 | 19.9754 | −59.9261 | 4.9938 |
| U8 | 0 | 0 | 0 | 0 | 0 | 0 | 0 | 0 | 0 | 0 | −479.409 | −59.926 | 479.4089 | −59.926 |
| θ8 | 0 | 0 | 0 | 0 | 0 | 0 | 0 | 0 | 0 | 0 | 59.9261 | 4.9938 | −59.9261 | 9.9877 |

[1] Notice the state sequence alternates the translation, and then rotation at each node.

**Table A2.** Mass matrix (M). [1]

| Mass | W2 | θ2 | W3 | θ3 | W4 | θ4 | W5 | θ5 | U6 | θ6 | u7 | θ7 | u8 | θ8 |
|---|---|---|---|---|---|---|---|---|---|---|---|---|---|---|
| W2 | 0.4761 | 0.0000 | 0.0037 | −0.0002 | 0 | 0 | 0 | 0 | 0 | 0 | 0 | 0 | 0 | 0 |
| θ2 | 0.0000 | 0.0000 | 0.0002 | −0.0001 | 0 | 0 | 0 | 0 | 0 | 0 | 0 | 0 | 0 | 0 |
| W3 | 0.0037 | 0.0002 | 0.476 | 0.0000 | 0.0037 | −0.0002 | 0 | 0 | 0 | 0 | 0 | 0 | 0 | 0 |
| θ3 | −0.0002 | −0.0001 | 0.0000 | 0.0000 | 0.0002 | −0.0001 | 0 | 0 | 0 | 0 | 0 | 0 | 0 | 0 |
| W4 | 0 | 0 | 0.0037 | 0.0002 | 0.4761 | 0.0000 | 0.0037 | −0.0002 | 0 | 0 | 0 | 0 | 0 | 0 |
| θ4 | 0 | 0 | −0.0002 | −0.0001 | 0.0000 | 0.0000 | 0.0002 | −0.0001 | 0 | 0 | 0 | 0 | 0 | 0 |
| W5 | 0 | 0 | 0 | 0 | 0.0037 | 0.0002 | 2.63 | −0.0004 | 0 | 0 | 0 | 0 | 0 | 0 |
| θ5 | 0 | 0 | 0 | 0 | −0.0002 | −0.0001 | −0.0004 | 0.0000 | 0.0002 | −0.0001 | 0 | 0 | 0 | 0 |
| U6 | 0 | 0 | 0 | 0 | 0 | 0 | 0 | 0.0002 | 0.476 | 0.0000 | 0.0037 | −0.0002 | 0 | 0 |
| θ6 | 0 | 0 | 0 | 0 | 0 | 0 | 0 | −0.0001 | 0.00 | 0.0000 | 0.0002 | −0.0001 | 0 | 0 |
| U7 | 0 | 0 | 0 | 0 | 0 | 0 | 0 | 0 | 0.0037 | 0.0002 | 0.4761 | 0.0000 | 0.0037 | −0.0002 |
| θ7 | 0 | 0 | 0 | 0 | 0 | 0 | 0 | 0 | −0.0002 | −0.0001 | 0.0000 | 0.0000 | 0.0002 | −0.0001 |
| U8 | 0 | 0 | 0 | 0 | 0 | 0 | 0 | 0 | 0 | 0 | 0.0037 | 0.0002 | 0.466 | −0.0004 |
| θ8 | 0 | 0 | 0 | 0 | 0 | 0 | 0 | 0 | 0 | 0 | −0.0002 | −0.0001 | −0.0004 | 0.0000 |

[1] Notice the state sequence alternates the translation, and then the rotation at each node.

**Table A3.** Natural frequencies, $\omega_n$ (rad/s). [1]

| 1809.5 | 1415.5 | 1042.2 | 774.3 | 596.8 |
|---|---|---|---|---|
| 478.8 | 410.9 | 54.9 | 43.7 | 30.9 |
| 15.8 | 10.2 | 0.7 | 2.1 | |

[1] Corresponding to the mode shapes in Table 5.

*Equations of Motion in Standard State Space Form*

$$\left\{ \begin{array}{c} \dot{x} \\ \ddot{x} \end{array} \right\}_{[nx1]} = [A]_{nxn}\left\{ \begin{array}{c} x \\ \dot{x} \end{array} \right\}_{nx1} + [B]_{nx1}\{u\}_{1x1} \tag{A6}$$

**Table A4.** Normalized mode shapes ($\times 10^4$).

| | | | | | | | | | | | | | |
|---|---|---|---|---|---|---|---|---|---|---|---|---|---|
| −1 | 3 | 2 | 0 | −3 | −5 | 3 | 1501 | 383 | 1037 | 443 | −692 | 181 | 240 |
| −1097 | 3080 | 4481 | 4992 | 4505 | 3173 | −1154 | −4544 | −136 | 2388 | 2221 | 4049 | 1395 | 1669 |
| −1 | 1 | −3 | −7 | −5 | 1 | −2 | −1569 | −215 | 204 | 667 | 1425 | −670 | −712 |
| −2158 | 4857 | 3958 | 28 | −3814 | −4883 | 2208 | −943 | −2040 | −7064 | −867 | 912 | −2460 | −1864 |
| −1 | −1 | −5 | 0 | 7 | 3 | 2 | 1296 | −125 | −1076 | 125 | 995 | −1385 | −1057 |
| −3111 | 4495 | −1058 | −4992 | −1185 | 4481 | −3142 | 5806 | 2118 | 368 | −2572 | 4061 | −3204 | −683 |
| 0 | 0 | 0 | 1 | 1 | −1 | 1 | −99 | 30 | 113 | −105 | 248 | 2247 | 954 |
| −3902 | 2199 | −4861 | −47 | 4875 | −2071 | 3937 | −4288 | −3426 | 4504 | 1878 | 4753 | −3652 | 1689 |
| 0 | −3 | 4 | 0 | −6 | 6 | 1 | 536 | −918 | 135 | 898 | 754 | −946 | 736 |
| −4493 | −1062 | −3138 | 4985 | −3062 | −1232 | −4519 | 815 | 2292 | −2423 | 3091 | 891 | −3893 | 3998 |
| 0 | 2 | 5 | −7 | 7 | −4 | 0 | −294 | 753 | 446 | 880 | 410 | −1936 | 1905 |
| −4872 | −3903 | 2129 | 75 | −2232 | 3920 | 4832 | −2471 | 3385 | −210 | −3658 | 3392 | −4013 | 5178 |
| −1 | −2 | 2 | −3 | 4 | −5 | −6 | 90 | −261 | 192 | −699 | −732 | −2945 | 3254 |
| −5031 | −5052 | 5012 | −5030 | 5062 | −4984 | −4965 | 3580 | −7823 | 3941 | −7649 | −5153 | −4046 | 5502 |

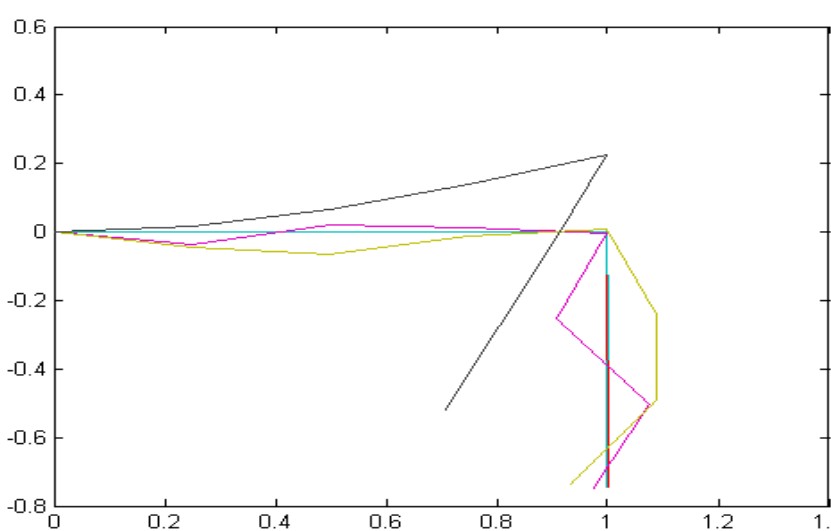

**Figure A1.** Normalized mode shapes (modal coordinates) displayed in physical coordinates with normalized length on the abscissa in meters and displacements in meters on the ordinate.

Finite element analysis performed in MATLAB (code is included in the Appendix) generates the mode shapes used to calculate the rigid-elastic coupling terms. The program outputs the flexible system $[A]$, $[B]$, $[C]$, and $[D]$ matrices of the standard state space form. The results are:

$$
\left\{ \begin{array}{c} \dot{x} \\ \ddot{x} \end{array} \right\}_{[nx1]} = \left[ \begin{array}{ccccc} 0 & 0 & 1 & 0 & \\ 0 & 0 & 0 & 1 & \\ \dfrac{-\omega_i^2 I_{zz}}{I_{zz}-D_i \sum_{i=1}^{n} D_i} & 0 & \dfrac{-2\xi\omega_i I_{zz}}{I_{zz}-D_i \sum_{i=1}^{n} D_i} & 0 & \\ \dfrac{-\omega_i^2 \sum_{i=1}^{n} D_i}{I_{zz}-D_i \sum_{i=1}^{n} D_i} & 0 & \dfrac{-2\xi\omega_i I_{zz}}{I_{zz}-D_i \sum_{i=1}^{n} D_i} & 0 & \end{array} \right]_{nxn} \left\{ \begin{array}{c} x \\ \dot{x} \end{array} \right\}_{nx1} + \left[ \begin{array}{c} 0 \\ 0 \\ \dfrac{-D_i T}{I_{zz}-D_i \sum_{i=1}^{n} D_i} \\ \dfrac{T}{I_{zz}-D_i \sum_{i=1}^{n} D_i} \end{array} \right]_{nx1} \{u\}_{1x1} \quad \text{(A7)}
$$

$$C = 100000000000 \quad D = 0 \quad \text{(A8)}$$

Given these equations, the resultant state space matrices are:

$$[B] = \left\{ \begin{array}{cccccccccccc} 0 & 0 & 0 & 0 & 0 & 0 & 0.126794 & -0.19565 & 0.126794 & 0.032156 & 0.002952 & 0.27828 \end{array} \right\}^T \quad \text{(A9)}$$

$$C = 100000000000 \quad D = 0 \quad \text{(A10)}$$

**Table A5.** State Space (A) matrix. [1]

| | 1 | 2 | 3 | 4 | 5 | 6 | 7 | 8 | 9 | 10 | 11 | 12 |
|---|---|---|---|---|---|---|---|---|---|---|---|---|
| 1 | 0 | 0 | 0 | 0 | 0 | 0 | 1 | 0 | 0 | 0 | 0 | 0 |
| 2 | 0 | 0 | 0 | 0 | 0 | 0 | 0 | 1 | 0 | 0 | 0 | 0 |
| 3 | 0 | 0 | 0 | 0 | 0 | 0 | 0 | 0 | 1 | 0 | 0 | 0 |
| 4 | 0 | 0 | 0 | 0 | 0 | 0 | 0 | 0 | 0 | 1 | 0 | 0 |
| 5 | 0 | 0 | 0 | 0 | 0 | 0 | 0 | 0 | 0 | 0 | 1 | 0 |
| 6 | 0 | 0 | 0 | 0 | 0 | 0 | 0 | 0 | 0 | 0 | 0 | 1 |
| 7 | 0 | 0.099 | −1.064 | −3.382 | −0.736 | −26.635 | 0 | $1.392\times10^{-4}$ | $-5.066\times10^{-4}$ | $-3.298\times10^{-4}$ | $-4.662\times10^{-5}$ | $-8.609\times10^{-4}$ |
| 8 | 0 | −0.659 | 1.642 | 5.218 | 1.136 | 41.100 | 0 | $-9.264\times10^{-4}$ | $7.817\times10^{-4}$ | $5.088\times10^{-4}$ | $7.194\times10^{-5}$ | $1.328\times10^{-3}$ |
| 9 | 0 | 0.188 | −6.435 | −6.433 | −1.401 | −50.670 | 0 | $2.649\times10^{-4}$ | $-3.064\times10^{-3}$ | $-6.273\times10^{-4}$ | $-8.869\times10^{-5}$ | $-1.638\times10^{-3}$ |
| 10 | 0 | 0.025 | −0.270 | −106.024 | −0.187 | −6.755 | 0 | $3.531\times10^{-5}$ | $-1.285\times10^{-4}$ | $-1.034\times10^{-2}$ | $-1.182\times10^{-5}$ | $-2.183\times10^{-4}$ |
| 11 | 0 | 0.002 | −0.025 | −0.079 | −249.447 | −0.620 | 0 | $3.241\times10^{-6}$ | $-1.179\times10^{-5}$ | $-7.677\times10^{-6}$ | $-1.579\times10^{-2}$ | $-2.004\times10^{-5}$ |
| 12 | 0 | 0.022 | −0.233 | −0.742 | −0.162 | −962.998 | 0 | $3.055\times10^{-5}$ | $-1.112\times10^{-4}$ | $-7.237\times10^{-5}$ | $-1.023\times10^{-5}$ | $-3.113\times10^{-2}$ |

[1] Flexible states where base (rigid body) rotation is controlled.

**Appendix B.**

Initialization Function Callbacks for the Simulation.
clear all; close all; clc;
%This block of code establishes the properties of each beam element
a = 0.0254; b = 0.0016; L = 0.25;
E = 72*10^9; I = a*b^3/12;
Li = [12 6*L -12 6*L;6*L 4*L^2 -6*L 2*L^2;-12 -6*L 12 -6*L;6*L 2*L^2 -6*L 4*L^2];
k_beam = E*I/L^3*[Li];
rho_beam = 2.8*10^3; %Beam density kg/m^3
A_beam = a*b; %Beam cross sectional area
mb = rho_beam*A_beam; %Beam mass per unit length
%This block creates the empty stiffness matrix [k]
k = zeros(14,14);
% This block fills in the stiffness matrix components
% Row 1 components start at index = 1% Row 2 components start at index = 15
k(1,1) = k_beam(3,3) + k_beam(1,1); k(2,1) = k_beam(4,3) + k_beam(2,1);
k(1,2) = k_beam(3,4) + k_beam(1,2); k(2,2) = k_beam(4,4) + k_beam(2,2);
k(1,3) = k_beam(1,3); k(2,3) = k_beam(2,3);
k(1,4) = k_beam(1,4); k(2,4) = k_beam(2,4);
% Row 3 components start at index = 29% Row 4 components start at index = 43
k(3,1) = k_beam(3,1); k(4,1) = k_beam(4,1);
k(3,2) = k_beam(3,2); k(4,2) = k_beam(4,2);
k(3,3) = k_beam(3,3) + k_beam(1,1); k(4,3) = k_beam(4,3) + k_beam(2,1);
k(3,4) = k_beam(3,4) + k_beam(1,2); k(4,4) = k_beam(4,4) + k_beam(2,2);
k(3,5) = k_beam(1,3); k(4,5) = k_beam(2,3);
k(3,6) = k_beam(1,4); k(4,6) = k_beam(2,4);
% Row 5 components start at index = 59% Row 6 components start at index = 73
k(5,3) = k_beam(3,1); k(6,3) = k_beam(4,1);
k(5,4) = k_beam(3,2); k(6,4) = k_beam(4,2);
k(5,5) = k_beam(3,3) + k_beam(1,1); k(6,5) = k_beam(4,3) + k_beam(2,1);
k(5,6) = k_beam(3,4) + k_beam(1,2); k(6,6) = k_beam(4,4) + k_beam(2,2);
k(5,7) = k_beam(1,3); k(6,7) = k_beam(2,3);
k(5,8) = k_beam(1,4); k(6,8) = k_beam(2,4);
% Row 7 components start at index = 89% Row 8 components start at index = 103
k(7,5) = k_beam(3,1); k(8,5) = k_beam(4,1);
k(7,6) = k_beam(3,2); k(8,6) = k_beam(4,2);
k(7,7) = k_beam(3,3); k(8,7) = k_beam(4,3);
k(7,8) = k_beam(3,4); k(8,8) = k_beam(4,4) + k_beam(2,2);
k(8,9) = k_beam(2,3);
k(8,10) = k_beam(2,4);
% Row 9 components start at index = 120% Row 10 components start at index = 134
k(9,8) = k_beam(3,2); k(10,8) = k_beam(4,2);

```
k(9,9) = k_beam(3,3) + k_beam(1,1); k(10,9) = k_beam(4,3) + k_beam(2,1);
k(9,10) = k_beam(3,4) + k_beam(1,2); k(10,10) = k_beam(4,4) + k_beam(2,2);
k(9,11) = k_beam(1,3); k(10,11) = k_beam(2,3);
k(9,12) = k_beam(1,4); k(10,12) = k_beam(2,4);
% Row 11 components start at index = 149% Row 12 components start at index = 163
k(11,9) = k_beam(3,1); k(12,9) = k_beam(4,1);
k(11,10) = k_beam(3,2); k(12,10) = k_beam(4,2);
k(11,11) = k_beam(3,3) + k_beam(1,1); k(12,11) = k_beam(4,3) + k_beam(2,1);
k(11,12) = k_beam(3,4) + k_beam(1,2); k(12,12) = k_beam(4,4) + k_beam(2,2);
k(11,13) = k_beam(1,3); k(12,13) = k_beam(2,3);
k(11,14) = k_beam(1,4); k(12,14) = k_beam(2,4);
% Row 13 components start at index = 179% Row 14 components start at index = 193
k(13,11) = k_beam(3,1); k(14,11) = k_beam(4,1);
k(13,12) = k_beam(3,2); k(14,12) = k_beam(4,2);
k(13,13) = k_beam(3,3); k(14,13) = k_beam(4,3);
k(13,14) = k_beam(3,4); k(14,14) = k_beam(4,4);
%Display stiffness matrix to check
k = k;
%END STIFFNESS MATRIX. START MASS MATRIX
%Assemble individual beam inertia matrix
I_beam = ones(1,8); %Creates empty matrix of I's for eight node points
I_beam = [I_beam.*I]; %Fill in matrix values with beam inertia
I_beam(1) = 0; %First node point inertia = 0
%This block of code creates the individual beam mass matrix "m_beam"
mi = [156 22*L 54 -13*L;22*L 4*L^2 13*L -3*L^2;54 13*L 156 -22*L;-13*L -3*L^2 -22*L 4*L^2];
m_beam = mb*L/420*mi;
%This block of code establishes the value of each point mass (mp)
%and the system point mass matrix (M)
mp = 0.455; %Point masses, M
M = [0 mp mp mp 2*mp mp mp mp]; %Matrix of 8 point masses (0 First point mass)
%Creates a 14 × 14 empty mass matrix [m]
m = zeros(14,14);
%Fill in the system mass matrix components
% Row 1 components start at index = 1% Row 2 components start at index = 15
m(1,1) = m_beam(3,3) + m_beam(1,1) + M(2); m(2,1) = m_beam(4,3) + m_beam(2,1);
m(1,2) = m_beam(3,4) + m_beam(1,2); m(2,2) = m_beam(4,4) + m_beam(2,2);
m(1,3) = m_beam(1,3); m(2,3) = m_beam(2,3);
m(1,4) = m_beam(1,4); m(2,4) = m_beam(2,4);
% Row 3 components start at index = 29% Row 4 components start at index = 43
m(3,1) = m_beam(3,1); m(4,1) = m_beam(4,1);
m(3,2) = m_beam(3,2); m(4,2) = m_beam(4,2);
m(3,3) = m_beam(3,3) + m_beam(1,1) + M(3); m(4,3) = m_beam(4,3) + m_beam(2,1);
m(3,4) = m_beam(3,4) + m_beam(1,2); m(4,4) = m_beam(4,4) + m_beam(2,2);
m(3,5) = m_beam(1,3); m(4,5) = m_beam(2,3);
m(3,6) = m_beam(1,4); m(4,6) = m_beam(2,4);
% Row 5 components start at index = 59% Row 6 components start at index = 73
m(5,3) = m_beam(3,1); m(6,3) = m_beam(4,1);
m(5,4) = m_beam(3,2); m(6,4) = m_beam(4,2);
m(5,5) = m_beam(3,3) + m_beam(1,1) + M(4); m(6,5) = m_beam(4,3) + m_beam(2,1);
m(5,6) = m_beam(3,4) + m_beam(1,2); m(6,6) = m_beam(4,4) + m_beam(2,2);
m(5,7) = m_beam(1,3); m(6,7) = m_beam(2,3);
m(5,8) = m_beam(1,4); m(6,8) = m_beam(2,4);
% Row 7 components start at index = 89
m(7,5) = m_beam(3,1);
```

```
m(7,6) = m_beam(3,2);
m(7,7) = m_beam(3,3) + 3*mb + M(5) + M(6) + M(7) + M(8);
m(7,8) = m_beam(3,4);
% Row 8 components start at index = 103% Row 9 components start at index = 120
m(8,5) = m_beam(4,1); m(9,8) = m_beam(3,2);
m(8,6) = m_beam(4,2); m(9,9) = m_beam(3,3) + m_beam(1,1) + M(6);
m(8,7) = m_beam(4,3); m(9,10) = m_beam(3,4) + m_beam(1,2);
m(8,8) = m_beam(4,4) + m_beam(2,2); m(9,11) = m_beam(1,3);
m(8,9) = m_beam(2,3); m(9,12) = m_beam(1,4);
m(8,10) = m_beam(2,4);
% Row 10 components start at index = 134% Row 11 components start at index = 149
m(10,8) = m_beam(4,2); m(11,9) = m_beam(3,1);
m(10,9) = m_beam(4,3) + m_beam(2,1); m(11,10) = m_beam(3,2);
m(10,10) = m_beam(4,4) + m_beam(2,2); m(11,11) = m_beam(3,3) + m_beam(1,1) + M(7);
m(10,11) = m_beam(2,3); m(11,12) = m_beam(3,4) + m_beam(1,2);
m(10,12) = m_beam(2,4); m(11,13) = m_beam(1,3);
m(11,14) = m_beam(1,4);
% Row 12 components start at index = 163
m(12,9) = m_beam(4,1);
m(12,10) = m_beam(4,2);
m(12,11) = m_beam(4,3) + m_beam(2,1);
m(12,12) = m_beam(4,4) + m_beam(2,2);
m(12,13) = m_beam(2,3);
m(12,14) = m_beam(2,4);
% Row 13 components start at index = 179% Row 14 components start at index = 193
m(13,11) = m_beam(3,1); m(14,11) = m_beam(4,1);
m(13,12) = m_beam(3,2); m(14,12) = m_beam(4,2);
m(13,13) = m_beam(3,3) + M(8); m(14,13) = m_beam(4,3);
m(13,14) = m_beam(3,4); m(14,14) = m_beam(4,4);
%Display the system mass matrix to check
m = m;
%Calculate the natural frequencies and normal modes
[NormalModes,EigenValues] = eig(inv(m)*k);
NaturalFrequencies = diag(EigenValues^0.5);
ModeShapes = NormalModes;
%Check Orthogonality like Homework 1 confirm diagonal matrix of 1's
%to satisfy Equation 24 on slide 17
OrthoMass = NormalModes'*m*NormalModes;
OrthoStiff = NormalModes'*k*NormalModes;
StiffCheck = OrthoStiff/EigenValues;
Equation24_OrthoCheck = diag(diag(StiffCheck/OrthoMass));
%Spacecraft Radius to be used designating rigid modal coordinate
R = 0.381;
FeeE = NormalModes; %Designate Elastic mode shapes array FeeE
Omega = NaturalFrequencies; %Designate variable name 'Omega' as natural frequencies
%Designate Rigid modal coordinate FeeR
FeeR = [R + L 1 R + L*2 1 R + L*3 1 R + L*4 1 -L 1 -L*2 1 -L*3 1];
Di = FeeE'*m*diag(FeeR); %Calculate Rigid-Elastic Coupling Coefficient
DiCheck = det(Di); %Confirm Di is singular...det(Di = 0)
Z = 0.0005;
Izz = 14;
w = diag(NaturalFrequencies); %Generate a diagonal matrix of natural frequency
Iw = 0.0912;
Td = 0; %Disturbance Torque
```

```
Tc = 0.1; %Control Torque is Iw*qddot_wheel
T = Td + Tc; %Total Torque is sum of disturbance and control torques
%Start State Space Development
NatFreq = diag(EigenValues).^0.5;
r = 0.381; %Radius of the wheel (large rigid body)
freqs = sqrt(EigenValues); %
NatFreq = EigenValues(1:5,1:5);
freqs = freqs(1:5,1:5);
zeta = 0.0005; %Given damping ratio for all modes
Izz = 14;
phi_E = NormalModes(1:14,1:5);
phi_R = [r + L,1,r + 2*L,1,r + 3*L,1,r + 4*L,1,-L,1,-2*L,1,-3*L,1]';
M_II = m;
Di = [phi_E'*M_II*phi_R];
M_state = [Izz Di';
Di eye(5)];
C_damp = [zeros(6,6)];
C_damp(2:6,2:6) = 2*zeta*freqs;
K = [zeros(6,6)];
K(2:6,2:6) = NatFreq;
A = [zeros(6),eye(6,6);
-inv(M_state)*K, -inv(M_state)*C_damp];
Bprime = [1;0;0;0;0;0];
B = [0 0 0 0 0 0 (inv(M_state)*Bprime)']';
C = zeros(12,12); C(1,1) = 1;
D = zeros(12,1);
[Gnum,Gden] = ss2tf(A,B,C,D);
G1 = tf(Gnum(1,:),Gden)
%Manually input Transfer Function to check
NUM = [1.998e-015 0.1268 0.007582 166.9 5.591 4.718e004 771 3.412e006 1.218e004
1.576e007 1.475e004 7.11e006];
DEN = [1 0.06125 1326 46.15 3.781e005 6683 2.808e007 1.388e005 1.813e008 2.065e005
9.954e007 0 0];
G = tf(NUM,DEN);
%Put PID controller Transfer function into workspace
It = 14; Z = 0.516931;
Bandwidth = 4; wn = Bandwidth; T = 10/Z/wn;
Kd = 2*Z*wn*It + It/T;
Kp = wn^2 + 2*Z*wn/T;
Ki = wn^2/T;
PID = tf([Kd Kp Ki],[0 1 0]);
%DESIGN FILTERS TO SMOOTH OUT MODE 1
%Design Bandpass filter for w = 10^-0.1478 = 0.711541 Hz
wz = 0.711541;Zz = 0.1;wp = wz;Zp = 0.0005;
BP1 = tf([1/wz^2 2*Zz/wz 1],[1/wp^2 2*Zp/wp 1]);
PID_BP1 = PID*BP1;
%Design Notch filter for w = 10^-0.109 = 0.778037 Hz
wz = 0.778037;Zz = 0.0005;wp = wz;Zp = 0.1;
Notch1 = tf([1/wz^2 2*Zz/wz 1],[1/wp^2 2*Zp/wp 1]);
Mode_1 = PID*BP1*Notch1;
%DESIGN FILTERS TO SMOOTH OUT MODE 2
%Design Bandpass filter for w = 10^0.3223
wz = 10^0.3223;Zz = 0.1;wp = wz;Zp = 0.0005;
BP2 = tf([1/wz^2 2*Zz/wz 1],[1/wp^2 2*Zp/wp 1]);
```

%Design Notch filter for w = 10^0.405
wz = 10^0.405;Zz = 0.0006;wp = wz;Zp = 0.1;
Notch2 = tf([1/wz^2 2*Zz/wz 1],[1/wp^2 2*Zp/wp 1]);
Mode_2 = Mode_1*BP2*Notch2;
%Design Lead filter for wz~1, wp~3
%wz = 1;Zz = 1;wp = 3;Zp = 1;
%Lead = tf([1/wz^2 2*Zz/wz 1],[1/wp^2 2*Zp/wp 1]);
%Mode_2 = Mode_2*Lead;
%DESIGN FILTERS TO SMOOTH OUT MODE 3
%Design Bandpass filter for w = 10^1.0110
wz = 10^1.0110;Zz = 0.1;wp = wz;Zp = 0.0005;
BP3 = tf([1/wz^2 2*Zz/wz 1],[1/wp^2 2*Zp/wp 1]);
%Design Notch filter for w = 10^1.0128
wz = 10^1.0128;Zz = 0.0005;wp = wz;Zp = 0.1;
Notch3 = tf([1/wz^2 2*Zz/wz 1],[1/wp^2 2*Zp/wp 1]);
Mode_3 = Mode_2*BP3*Notch3;
%DESIGN FILTERS TO SMOOTH OUT MODE 4
%Design Bandpass filter for w = 10^1.49035
wz = 10^1.49035;Zz = 0.1;wp = wz;Zp = 0.0005;
BP4 = tf([1/wz^2 2*Zz/wz 1],[1/wp^2 2*Zp/wp 1]);
%Design Notch filter for w = 10^1.492
wz = 10^1.492;Zz = 0.0005;wp = wz;Zp = 0.1;
Notch4 = tf([1/wz^2 2*Zz/wz 1],[1/wp^2 2*Zp/wp 1]);
Mode_4 = Mode_3*BP4*Notch4;
%CALCULATE SYSTEM NATURAL FREQUENCIES
[NaturalFrequencies,Damping,EigenValue] = damp(G);
NaturalFrequencies = NaturalFrequencies;

**Appendix C.**

Stop Function Callbacks for the Simulation.
[mag1,phase1,wout1] = bode(G); Mag1 = 20*log10(mag1(:)); Phase1 = phase1(:);
[mag2,phase2,wout2] = bode(G*PID); Mag2 = 20*log10(mag2(:)); Phase2 = phase2(:);
[mag3,phase3,wout3] = bode(G*PID*BP1); Mag3 = 20*log10(mag3(:)); Phase3 = phase3(:);
[mag4,phase4,wout4] = bode(G*PID*BP1*Notch1); Mag4 = 20*log10(mag4(:));
Phase4 = phase4(:);
[mag5,phase5,wout5] = bode(G*PID*BP1*Notch1*BP2); Mag5 = 20*log10(mag5(:));
Phase5 = phase5(:);
[mag6,phase6,wout6] = bode(G*PID*BP1*Notch1*BP2*Notch2); Mag6 = 20*log10
(mag6(:)); Phase6 = phase6(:);
[mag7,phase7,wout7] = bode(G*PID*BP1*Notch1*BP2*Notch2*BP3); Mag7 = 20*log10
(mag7(:)); Phase7 = phase7(:);
[mag8,phase8,wout8]  =  bode(G*PID*BP1*Notch1*BP2*Notch2*BP3*Notch3);
Mag8 = 20*log10(mag8(:)); Phase8 = phase8(:);
[mag9,phase9,wout9] = bode(G*PID*BP1*Notch1*BP2*Notch2*BP3*Notch3*BP4);
Mag9 = 20*log10(mag9(:)); Phase9 = phase9(:);
[mag10,phase10,wout10] = bode(G*PID*BP1*Notch1*BP2*Notch2*BP3*Notch3*BP4*
Notch4); Mag10 = 20*log10(mag10(:)); Phase10 = phase10(:);
figure(1); hold on;
semilogx(wout1,Mag1,'−','LineWidth',1);
semilogx(wout2,Mag2,'LineWidth',1);
semilogx(wout3,Mag3,'−','LineWidth',3);
semilogx(wout4,Mag4,':','LineWidth',2);
hold off; grid on; axis([0.5,40, −100, 150]); set(gca, 'FontSize',28, 'FontName',
'Palatino Linotype');

```
legend('Flexible space robot','PID','PID + Bandpass','PID + Notch + Bandpass')
figure(2); hold on; set(gca, 'FontSize',28, 'FontName','Palatino Linotype');
semilogx(wout1,Phase1,'−','LineWidth',1);
semilogx(wout2,Phase2,'LineWidth',1);
semilogx(wout3,Phase3,'−','LineWidth',3);
semilogx(wout4,Phase4,':','LineWidth',2);
hold off; grid on;
figure(3); hold on;
semilogx(wout2,Mag2,'−','LineWidth',1);
semilogx(wout4,Mag4,'LineWidth',1);
semilogx(wout5,Mag5,'−','LineWidth',3);
semilogx(wout6,Mag6,':','LineWidth',2);
hold off; grid on; axis([0.5,40, −100, 150]); set(gca, 'FontSize',28, 'FontName',
'Palatino Linotype');
legend('PID controlled Flexible space robot','PID + Mode 1','PID + Mode 1 + Band-
pass','PID + Mode 1 + Notch + Bandpass')
figure(4); hold on; set(gca, 'FontSize',28, 'FontName','Palatino Linotype');
semilogx(wout2,Phase2,'−','LineWidth',1);
semilogx(wout4,Phase4,'LineWidth',1);
semilogx(wout5,Phase5,'−','LineWidth',3);
semilogx(wout6,Phase6,':','LineWidth',2);
hold off; grid on;
figure(5); hold on;
semilogx(wout2,Mag2,'−','LineWidth',1);
semilogx(wout4,Mag4,'LineWidth',1);
semilogx(wout6,Mag6,'−','LineWidth',3);
semilogx(wout7,Mag7,':','LineWidth',2);
semilogx(wout8,Mag8,':','LineWidth',2);
hold off; grid on; axis([0.5,40, −100, 150]); set(gca, 'FontSize',28, 'FontName',
'Palatino Linotype');
legend('PID controlled Flexible space robot','PID + Mode 1','PID + Mode 2','PID +
Mode 1 + Mode 2 + Bandpass','PID + Mode 1 + Mode 2 + Bandpass + Notch')
figure(6); hold on; set(gca, 'FontSize',28, 'FontName','Palatino Linotype');
semilogx(wout2,Phase2,'−','LineWidth',1);
semilogx(wout4,Phase4,'LineWidth',1);
semilogx(wout6,Phase6,'−','LineWidth',3);
semilogx(wout7,Phase7,':','LineWidth',2);
semilogx(wout8,Phase8,':','LineWidth',2);
hold off; grid on;
figure(7); hold on;
semilogx(wout2,Mag2,'−','LineWidth',1);
semilogx(wout4,Mag4,'LineWidth',1);
semilogx(wout6,Mag6,'−','LineWidth',3);
semilogx(wout8,Mag8,':','LineWidth',2);
semilogx(wout9,Mag9,':','LineWidth',2);
semilogx(wout10,Mag10,':','LineWidth',2);
hold off; grid on; axis([0.5,40, −100, 150]); set(gca, 'FontSize',28, 'FontName',
'Palatino Linotype');
legend('PID controlled Flexible space robot','PID + Mode 1','PID + Mode 2','PID +
Mode 1 + Mode 2 + Mode 3','PID + Mode 1 + Mode 2 + Mode 3 + Bandpass + Notch')
figure(8); hold on; set(gca, 'FontSize',28, 'FontName','Palatino Linotype');
semilogx(wout2,Phase2,'−','LineWidth',1);
semilogx(wout4,Phase4,'LineWidth',1);
semilogx(wout6,Phase6,'−','LineWidth',3);
```

```
semilogx(wout8,Phase8,':','LineWidth',2);
semilogx(wout9,Phase9,':','LineWidth',2);
semilogx(wout10,Phase10,':','LineWidth',2);
hold off; grid on;
sys1 = G*PID/(1 + G*PID);
sys2 = (G*PID*BP1/(1 + G*PID*BP1));
sys3 = (G*PID*BP1*Notch1/(1 + G*PID*BP1*Notch1));
sys4 = (G*PID*BP1*Notch1*BP2/(1 + G*PID*BP1*Notch1*BP2));
sys5 = (G*PID*BP1*Notch1*BP2*Notch2/(1 + G*PID*BP1*Notch1*BP2*Notch2));
sys6 = (G*PID*BP1*Notch1*BP2*Notch2*BP3/(1 + G*PID*BP1*Notch1*BP2*Notch2*BP3));
sys7 = (G*PID*BP1*Notch1*BP2*Notch2*BP3*Notch3/(1 + G*PID*BP1*Notch1*BP2*Notch2*BP3*Notch3));
sys8 = (G*PID*BP1*Notch1*BP2*Notch2*BP3*Notch3*BP4/(1 + G*PID*BP1*Notch1*BP2*Notch2*BP3*Notch3*BP4));
sys9 = (G*PID*BP1*Notch1*BP2*Notch2*BP3*Notch3*BP4*Notch4/(1 + G*PID*BP1*Notch1*BP2*Notch2*BP3*Notch3*BP4*Notch4));
figure(9); step(sys1,sys2); legend('PID','PID + BP1');set(gca, 'FontSize',28, 'FontName','Palatino Linotype');
figure(10); step(sys1,sys3); legend('PID','PID + BP1 + Notch1'); set(gca, 'FontSize',28, 'FontName','Palatino Linotype');
figure(11); step(sys1,sys4); legend('PID','PID + Mode 1 + BP2'); set(gca, 'FontSize',28, 'FontName','Palatino Linotype');
figure(12); step(sys1,sys5); legend('PID','PID + Mode 1 + BP2 + Notch 2'); set(gca, 'FontSize',28, 'FontName','Palatino Linotype');
figure(13); step(sys1,sys6); legend('PID','PID + Mode 1 + Mode 2 + BP3'); set(gca, 'FontSize',28, 'FontName','Palatino Linotype');
figure(14); step(sys1,sys7); legend('PID','PID + Mode 1 + Mode 2 + BP3 + Notch3'); set(gca, 'FontSize',28, 'FontName','Palatino Linotype');
figure(15); step(sys1,sys8); legend('PID','PID + Mode 1 + Mode 2 + Mode 3 + BP4'); set(gca, 'FontSize',28, 'FontName','Palatino Linotype');
figure(16); step(sys1,sys9); legend('PID','PID + Mode 1 + Mode 2 + Mode 3 + BP4 + Notch4'); set(gca, 'FontSize',28, 'FontName','Palatino Linotype');
```

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
