# Peer review of "Flattening the Curve of Flexible Space Robotics"

_applsci, doi:10.3390/app12062992_

Round 1
Reviewer 1 Report
The manuscript deals with flattening the curve of flexible space robotics.
The manuscript is well presented and the manuscript contribution is well stated.
Author Response
Thank you for your favorable review. The worthiness of the contribution is appreciated, but more so appreciated is the compliments on presentation. Prior to submission, the manuscript underwent significant review and improvement by writing professionals whose efforts are manifestly apparent in the presentation quality.
Reviewer 2 Report
The paper surveys the control methods that can be applied in flexible robotics, particularly for application in space and compares them in numerical simulations.Comments:
- The historical part of the paper is very long but not obligatorily /necessary. This could be shortened with the citation of some relevant literature that covers some fields already (e.g., Xu, Y., & Kanade, T. (Eds.). (2012). Space robotics: dynamics and control (Vol. 188). Springer Science & Business Media., , https://doi.org/10.1016/j.icarus.2021.114558, doi: 10.1109/IROS.2012.6386169, https://doi.org/10.1016/j.actaastro.2012.06.010 )
Other parts of the paper should also be condensed, and more computational parts should be moved to the appendix. Minor comments:
- The paper shouldn't refer to itself as a "manuscript".
- In equation (7), "V" denotes potential energy and linear velocity too.
- In equation (4), "J" is used but not explained.
- In equation (5), "J" is used in another meaning without explanation.
- In equations (37-38), omega_1 and omega_2 are not explained.
- Figure 11: consider redrawing it instead of copying a low-quality image
- There is much trivial information as
- software-related images, as Figure 13/b,
- subsection 2.13.
- Name the curves of Figure 13/a (preferably using standard legend box)
- Figure 15: why are necessary the small plots?
Author Response
Thank you for your favorable review and thanks for your time producing a detailed review.
- The historical part of the paper is very long but not obligatorily /necessary. This could be shortened with the citation of some relevant literature that covers some fields already
- The reviewer recommends two additional citations to articles by Tamás Haidegger pertaining to medical space robotics. The motivation for low-latency telepresence in the context of robotic surgery in space is appreciated as an enhancement to the literature review. While not intended in the research’s original scope, the recommended citations enhance the readership’s understanding of the broad applicability of the research.
- Section 2 has been shortened eliminating the ties to first sources (historical parts) with additional shortening by elimination of references only cited as historical ties from the seminar literature to the present to satisfy the requested revision.
- Other parts of the paper should also be condensed, and more computational parts should be moved to the appendix.
- Input shaping, piezo-electrics, and sensor/actuator placement sections have all been eliminated completely, since the methods was merely included for context, but not implemented in comparison to the proposed methods. The manuscript is 9% shorter by word count and 5% by page count.
- The paper shouldn't refer to itself as a "manuscript".
- The descriptor “manuscript” has been eliminated in fourteen instances.
- In equation (7), "V" denotes potential energy and linear velocity too.
- Similarly, T is retained for kinetic energy where wheel torques is modified to in equations (4), (15)-(19), (24)-(27) and (41). V is retained as potential energy, while linear velocity is represented by in line 234, in line 306, in line 307, in line 308
- In equation (4), "J" is used but not explained.
- The term is now defined in the list following the equation.
- In equation (5), "J" is used in another meaning without explanation.
- The cost functional is equation (5) is no longer represented in variable form
- In equations (37-38), omega_1 and omega_2 are not explained.
- Great catch. The explanation has been added in the text immediately prior to the two equations.
- Figure 11: consider redrawing it instead of copying a low quality image
- Figure 11 is an original, manually created graphic drawn by the author mimicking the cited reference, rather than a low quality copied image, but the image did not meet the MDPI guidance stipulating preference for minimum resolution of 300 dpi or higher. Images have been re-saved at 300 dpi and inserted in the revised manuscript (the line thicknesses are intentional ?).
- There is much trivial information as software-related images, as Figure 13/b, subsection 2.13.
- Figure 13.b is a topological image taken directly from implementation in SIMULINK and is key to repeatability of the results by the readership. Figure 13.b indicates that no other changes were made other than flexible robotic accommodation methodology, evidenced by the manual signal switches. Section 2.13 comprises two sentences stipulating the software version and settings used to produce the results presented in the reviewed manuscript, and inclusion of such is deemed a “best practice” to aid repeatability by the readership.
- Name the curves of Figure 13/a (preferably using standard legend box)
- Thanks for the catch. The names of these curves are the key ties to the article’s title. Names of curves are added to the figure caption.
- Figure 15: why are necessary the small plots?
- The author agrees with the reviewer that the new MDPI document template eliminating one third of the document-width provides challenges. In the reviewed (lengthy) manuscript, special attention has been paid to present the plotted data in minimal space, and the threshold size has been established by ensuring the internal content is discernable while the internal font size is roughly equivalent to the figure caption (the smallest font size permissible in the template). An additional consideration is the desire to increase readability of the article by placing “mini-tables” as subfigures accompanying qualitative plots, so the readers are not required to flip back and forth between pages to connect the qualitative plots with the quantitative tables.
Reviewer 3 Report
In relation to the format, numerous figures and tables that exceed the margins are detected. Also, the text contains some paragraph in italics apparently by mistake.
The contents seem adequate.
Author Response
Thank you for your favorable review. The adequacy of the content is well received. The use of italics for emphasis is admittedly odd for technical manuscripts, but useful application of “literary license” to aid the readership understanding the very lengthy manuscript.
Reviewer 4 Report
Excellent work. It deserves to be published. Your findings might indicate that the system wants to be driven from initial state to the terminal state by a function with minimum curvature, or with the minimal integral of squared acceleration. A sinusoidal is near the optimum. The optimum is given by a cubic polynomial, cf. Hint.pdf (Maple worksheet).

Author Response
Thank you for your favorable review and assertion of publish-ability. The reviewer’s discussion of minimum curvature or minimal integral of the squared acceleration is intriguing and has been added as a logical next step in the research lineage in section 4.2, especially in light of sinusoidal approximation of cubic polynomials.